# Multi-omics analysis of hospital-acquired diarrhoeal patients reveals biomarkers of enterococcal proliferation and *Clostridioides difficile* infection

Marijana Bosnjak [1], Avinash V. Karpe[2,3,4], Thi Thu Hao Van[5], Despina Kotsanas[4], Grant A. Jenkin[6], Samuel P. Costello[7], Priscilla Johanesen [1], Robert J. Moore [5], David J. Beale[2], Yogitha N. Srikhanta[1], Enzo A. Palombo [3], Sarah Larcombe[1,8] & Dena Lyras [1,8] ✉

Hospital-acquired diarrhoea (HAD) is common, and often associated with gut microbiota and metabolome dysbiosis following antibiotic administration. *Clostridioides difficile* is the most significant antibiotic-associated diarrhoeal (AAD) pathogen, but less is known about the microbiota and metabolome associated with AAD and *C. difficile* infection (CDI) with contrasting antibiotic treatment. We characterised faecal microbiota and metabolome for 169 HAD patients (33 with CDI and 133 non-CDI) to determine dysbiosis biomarkers and gain insights into metabolic strategies *C. difficile* might use for gut colonisation. The specimen microbial community was analysed using 16 S rRNA gene amplicon sequencing, coupled with untargeted metabolite profiling using gas chromatography-mass spectrometry (GC-MS), and short-chain fatty acid (SCFA) profiling using GC-MS. AAD and CDI patients were associated with a spectrum of dysbiosis reflecting non-antibiotic, short-term, and extended-antibiotic treatment. Notably, extended antibiotic treatment was associated with enterococcal proliferation (mostly vancomycin-resistant *Enterococcus faecium*) coupled with putative biomarkers of enterococcal tyrosine decarboxylation. We also uncovered unrecognised metabolome dynamics associated with concomitant enterococcal proliferation and CDI, including biomarkers of Stickland fermentation and amino acid competition that could distinguish CDI from non−CDI patients. Here we show, candidate metabolic biomarkers for diagnostic development with possible implications for CDI and vancomycin-resistant enterococci (VRE) treatment.

Hospital-acquired diarrhoea (HAD) is an acute diarrhoeal episode that arises after ≥3 days of hospitalisation and is common, particularly among elderly patients[1]. HAD is most often associated with non-infective factors, notably antibiotic administration, that perturb gut microbiota[2]. Antibiotic exposure is a significant risk factor for developing *Clostridioides difficile* infection (CDI), which accounts for up to 15-25% of HAD cases and is characterised by gastrointestinal inflammation resulting in mild-to-severe diarrhoea, pseudomembranous

colitis, toxic megacolon, and in severe cases, death[3–5]. Because of shared risk factors, including combination and extended antibiotic treatment, the microbiota of antibiotic-associated diarrhoea (AAD) and CDI patients are also associated with the proliferation of vancomycin-resistant enterococci (VRE) that can result in poor patient outcomes[6,7]. Despite antibiotic risks, clinically, the microbiota and metabolome profiles for AAD, CDI and concomitant VRE have been poorly defined.

Previous studies have characterised the AAD and CDI microbiota as heterogeneous in composition but reduced in bacterial diversity and richness, often with concomitant Enterobacteriaceae or *Enterococcus* proliferation[8,9]. Microbiota differences between CDI and non-CDI patients have been attributed to the loss of putatively protective genera or increases in mucin-degrading genera such as *Akkermansia*[10,11]. Clinical studies accounting for contrasting antibiotic treatments in their assessment of CDI are rare. However, concomitant treatment with several antibiotic classes has been associated with decreased bacterial richness and high proportions of Enterobacteriaceae (mostly *Escherichia* spp.) among CDI patients[9–11]. Animal CDI models show that while the CDI microbiota reflects the effects of different antibiotic classes and exposure periods, as CDI progresses, microbiota compositional changes have been attributed directly to *C. difficile* toxin-mediated inflammation[12,13]. However, human studies have not directly detected these *C. difficile* microbiota-associated changes.

Enterococcal proliferation is associated with an increased risk of CDI[6,8] and has been detected in non-diarrhoeal hospitalised patients[14]. VRE proliferation may be partly driven by oral administration of antibiotics as the standard first-line therapy for CDI[15]. VRE proliferation is particularly significant in the hospital setting, which places vulnerable patients at increased risk of bacteraemia[6,16]. Decolonisation of VRE is difficult, and patients can remain colonised for prolonged periods, serving as reservoirs for transmission and infection to others. While concomitant CDI and VRE are associated with worse patient outcomes[7,17], little is known about how the metabolome may contribute to this worsening.

The loss of key commensals after antibiotic exposure alters bacterial metabolism and secondary metabolite production[12,13,18], with increasing evidence that the gut metabolome is vital in driving *C. difficile* growth, proliferation, and toxin production[12,13,17–20]. CDI risk and pathogenesis have been associated with elevated gastrointestinal primary bile acids, amino acids and SCFA depletion[20–22]. Primary bile acids have been shown to increase *C. difficile* spore germination and vegetative cell growth in vitro[20]. Microbiota-derived SCFAs, particularly butyrate, the primary energy source for intestinal epithelial cells (IEC), helped maintain intestinal barrier integrity and ameliorate toxin-mediated inflammation in CDI mice[23,24]. Furthermore, microbiota restoration through faecal microbiota transplantation (FMT) has been shown to restore secondary bile acid and butyrate concentrations, resolve CDI symptoms in patients, and decolonise *C. difficile*[21,25]. However, the role of SCFAs in CDI is less clear, as a recent study found that inoculating gnotobiotic CDI mice with butyrate-producing clostridia increased the relative abundance of butyrate but worsened disease progression[26], suggesting that *C. difficile* senses and responds to SCFAs and modulates virulence accordingly to maintain dysbiosis[24].

In vitro culture studies and in vivo mouse CDI models show that *C. difficile* metabolises many substrates, including sugars, sugar alcohols and amino acids. Elevated amino acids processed via Stickland reactions, particularly proline, are the preferred energy source that drives rapid bacterial growth and increases CDI susceptibility[27–29]. A recent mouse infection model found *Enterococcus* provided *C. difficile* with a source of amino acids that increased *C. difficile* pathogenesis[17]. Conversely, FMT restoration of commensal microbiota increased competition for these preferred amino acids where species such as *Clostridium sardiniense*, with

similar nutritional requirements as *C. difficile*, deplete amino acids in the gut to provide substantial protection against CDI[26]. Stickland products coupled with a decrease in amino acid substrates have been observed in in vivo CDI studies suggesting utilisation and vegetative growth[18]. However, in clinical studies, evidence of Stickland amino acid fermentation by-products has been variable. A recent clinical study found by-products of L-leucine fermentation rather than L-proline fermentation to be a putative biomarker of toxigenic CDI[30], but these observations did not consider the contributions of other clostridia competing with *C. difficile*.

We hypothesised that AAD and CDI associated with enterococcal proliferation and differences in antibiotic exposure might be associated with biomarkers that provide insights into microbiota and *C. difficile* metabolic strategies during infection. We conducted a retrospective study of 169 hospital-acquired diarrhoeal patients (33 with CDI) and presented detailed microbiota and metabolomics analyses using 16 S rRNA gene amplicon sequencing, untargeted and SCFA GC-MS-based metabolomic profiling. Univariate and multivariate modelling and statistical techniques were used to investigate how CDI and non-CDI microbiota and metabolome composition differed with variations in antibiotic treatment.

## Results

### Microbiota associations with extended antibiotic exposure

We first assessed HAD gut microbiota structure and diversity with respect to antibiotic treatment using alpha and beta diversity measures and analysis of compositions of microbiomes (ANCOM) differential abundance analysis. Antibiotic-associated diarrhoeal (+AAD) patients comprised 82.2% (139/169) of the cohort (Table 1) and were associated with 56 unique combinations of antibiotic classes before specimen collection (Supplementary Table 1), rendering analysis by antibiotic class unfeasible. For comparison, faecal samples from healthy donors recruited for faecal microbiota transplant treatment of CDI were similarly assessed (Supplementary materials). Donor samples in the study were obtained from 12 female and 8 male individuals, a smaller male cohort (40.0% male) compared to CDI patients (54.5% male) and non-CDI patients (47.4% male). FMT donors were between the ages of 18 and 65 46 with a total age of 649 years and an average of 32.45 years. While the average age of FMT donors was significantly lower than the CDI (75 years, range 55-83) and non-CDI (68 years, range 52-78) median age, we chose FMT donors purposefully as a comparison group for this study in order to assess the microbiota and metabolomes of CDI and non-CDI patients against FMT donors who are medically assessed as healthy and are actively recruited to treat recurrent and severe CDI. To analyse AAD microbiota variation, we generated two models that controlled for the period of antibiotic exposure and the number of antibiotic classes. AAD patients treated for different periods or with an increasing number of antibiotic classes did not yield significant between-group differences in alpha diversity (Fig. 1a, b). However, analysis of taxonomic composition showed that in total, 35% (61/169) of samples had an elevated abundance of *Enterococcus*, comprising 25-99% of the gut microbiota, and that the mean abundances of *Enterococcus* increased with extended periods of antibiotic exposure (≥3 days) and an increasing number of antibiotic classes (≥2 classes) (Fig. 1c, d). Enterococcal-dominant AAD specimens were cultured (see methods) and MALDI-TOF mass spectroscopy determined that the predominant species present in these samples was *E. faecium*. The final analysis was performed on 56 isolates with four removed due to poor quality sequences. The majority of *E. faecium* isolates identified (64%, 36/56) encoded vancomycin resistance determinants *vanA* or *vanB*, with 16% (9/56) encoding *vanA*, approximately 52% (29/56) encoding *vanB*, and approximately 3.5% (2/56) encoding both *vanA* and *vanB* (Fig. 2). Approximately 46% (26/56) of *E. faecium* isolates identified belonged to the epidemic ST796 clonal group, all of which encoded *vanB*, including the two isolates that

**Table 1 | Patient demographics and antibiotic usage**

| Subject characteristics | Non-CDI n = 136 | CDI n = 33 | Non-CDI vs CDI P-value | Statistical test |
|---|---|---|---|---|
| Gender (Male) % | 64 (47.4%) | 18 (54.5%) | 0.563 | Chi-square[6] |
| Age, median years (IQR) | 68 (52-78) | 75 (55-83) | 0.181 | Mann–Whitney U[7] |
| Days hospitalisation, median (IQR) | 8 (5-13) | 9 (6-14) | 0.528 | Mann–Whitney U |
| Prior hospitalisation[1], n (%) | 83 (60.6%) | 27 (81.8%) | 0.079 | Chi-square |
| Multiple hospitalisations[1], n (%) | 56 (41.2%) | 17 (51.5%) | 0.144 | Chi-square |
| Antibiotic and non-antibiotic medications | | | | |
| Any antibiotic, n (%) [2] | 116 (69.7%) | 23 (84.7%) | 0.084 | Chi-square |
| Days antibiotic treatment, median (IQR)[3] | 5 (3-9) | 4 (2-11) | 0.935 | Mann–Whitney U |
| Number of antibiotic classes, median (IQR)[4] | 2 (1–3) | 2 (1–3) | 0.835 | Mann–Whitney U |
| Any PPI, n (%) | 94 (68.6%) | 26 (78.8%) | 0.336 | Chi-square |
| PPI + antibiotics, n (%) | 81 (59.1%) | 19 (57.6%) | 0.139 | Chi-square |
| Chemotherapy, n (%) | 26 (19.0%) | 7 (21.2%) | 0.978 | Chi-square |
| Chemotherapy + antibiotics, n (%) | 22 (16.2%) | 4 (12.1%) | 0.823 | Chi-square |
| Antibiotic classes[2], n (%) | | | | |
| Aminoglycoside | 11 (9.5%) | 0 (0%) | 0.297 | Chi-square |
| Carbapenem | 13 (11.2%) | 0 (0%) | 0.224 | Chi-square |
| Cephalosporin | 60 (51.7%) | 16 (76.2%) | 0.061 | Chi-square |
| Cyclic lipopeptide | 3 (2.6%) | 0 (0%) | 1.000 | Chi-square |
| Fluoroquinolone | 9 (7.8%) | 3 (14.3%) | 0.337 | Chi-square |
| Glycopeptide | 15 (12.9%) | 2 (9.5%) | 0.939 | Chi-square |
| Clindamycin | 1 (0.9%) | 2 (9.5%) | 0.092 | Chi-square |
| Macrolide | 18 (15.5%) | 2 (9.5%) | 0.704 | Chi-square |
| Metronidazole | 33 (28.4%) | 6 (28.6%) | 1.000 | Chi-square |
| Penicillin | 19 (16.4%) | 2 (9.5%) | 0.636 | Chi-square |
| Penicillin β-lactamase (Oral) | 16 (13.8%) | 3 (14.3%) | 1.000 | Chi-square |
| Penicillin β-lactamase (IV) | 42 (36.2%) | 8 (38.1%) | 1.000 | Chi-square |
| Trimethoprim | 4 (3.4%) | 0 (0%) | 0.873 | Chi-square |
| Tetracycline | 6 (5.2%) | 1 (4.8%) | 1.000 | Chi-square |
| Other [5] | 1 (0.9%) | 1 (4.8%) | - | - |

[1]Hospitalisation in the 12 months preceding specimen collection
[2]n = 139 antibiotic-associated diarrhoeal (+AAD) patients
[3]n = 136 + AAD patients as antibiotic treatment data were missing for three patients
[4]n = 138 + AAD patients as antibiotic treatment data were missing for one patient
[5]Other (Rifampicin and Linezolid)
[6]2-sided significance
[7]2-sided significance

encoded both *vanA* and *vanB* (Fig. 2). The next most prevalent sequence types included ST18 (10.7%, 6/56), ST1421 (7.1%, 4/56), and ST203 (5.4%, 3/56).

**Enterococcal proliferation and low diversity microbiota**
To determine microbiota and metabolome associations with low diversity enterococcal-dominant AAD, we stratified HAD patients into non-antibiotic (-AAD), non-enterococcal dominant antibiotic-associated diarrhoea (-Ent AAD), and enterococcal-dominant antibiotic-associated diarrhoea ( + Ent AAD) groups (Fig. 3a). Only patients in which the *Enterococcus* 16 S rRNA gene amplicon sequences contributed ≥25% of the total microbiota were included in the +Ent AAD group.

+Ent AAD patients formed a microbially distinct subset of AAD characterised by lower diversity. Plots of alpha diversity measurements showed that compared to FMT donors, all diarrhoeal groups (-AAD, -Ent AAD and +Ent AAD) were associated with a considerable spread of alpha diversity values and significantly lower mean Shannon indices (all $p < 0.0001$) (Fig. 3b). While there was no significant difference between -AAD and -Ent AAD (p > 0.05), the mean Shannon index for +Ent AAD patients was significantly reduced compared to -AAD and

-Ent AAD patients (all $p < 0.0001$) (Fig. 3c). Furthermore, the ordination plot visualising Bray-Curtis dissimilarities highlighted that +AAD patients formed two distinct clusters, with -Ent AAD patients clustered left with -AAD patients while +Ent AAD patients clustered right (Fig. 3c). Pairwise post-hoc PERMANOVA determined that the difference in distribution of centroids for -AAD vs -Ent AAD was insignificant ($R^2 = 0.010$, p = 0.415), but significant for -Ent AAD vs +Ent AAD ($R^2 = 0.325$, $p < 0.0001$).

**Low diversity enterococcal AAD formed a metabolically distinct subset of AAD**
The heatmap of the mean abundance of these 97 metabolites that best described the variation in the enterococcal-dominance PLS-DA model revealed the +Ent AAD metabolome was elevated across several classes (alcohols, amines, amino acids, primary bile acids, and sugars) and depleted in indoles, fatty acids, and phenylpropanoic acids compared to FMT donors, -AAD and -Ent AAD patients (Supplementary Fig. 1A).

The PLS-DA scores plot in Supplementary Fig. 1B showed that while FMT donors separated from -AAD, -Ent AAD, and +Ent AAD patients, there was no clear separation between the diarrhoeal groups. However, +Ent AAD patients (yellow) clustered further

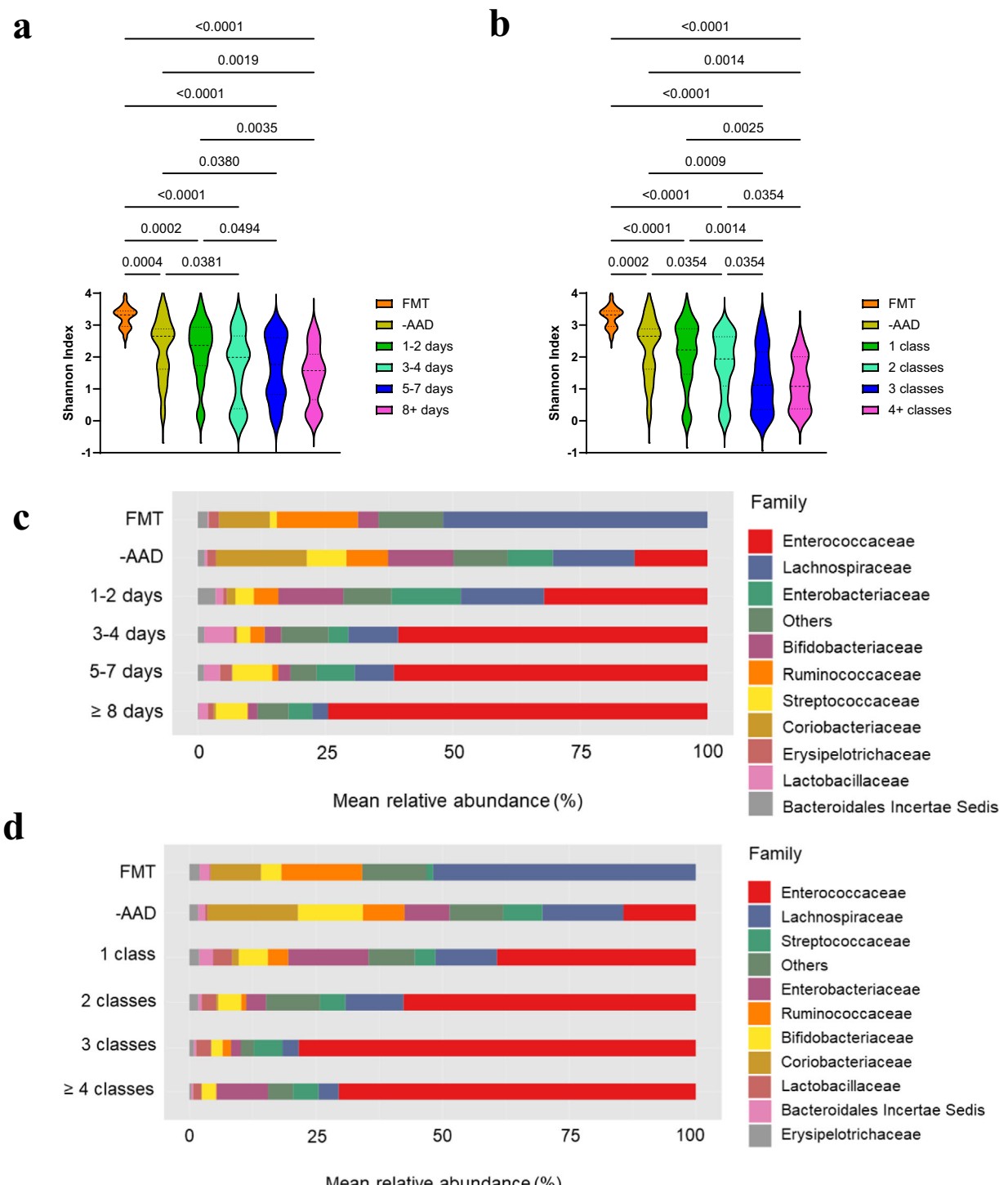

**Fig. 1 | Extended antibiotic exposure and combination antibiotic therapy was associated with microbiota dominated by Enterococcaceae.** Violin plots of Shannon diversity indices assessed species richness and evenness among **a** FMT donors (n = 20), non-antibiotic AAD (-AAD) (n = 29), 1–2 days (n = 29), 3–4 days (n = 36), 5–7 days (n = 28) and ≥ 8 days (n = 37) antibiotic treatment groups, and, **b** FMT donors (n = 20), non-antibiotic AAD (-AAD) (n = 29), 1 class (n = 49), 2 classes (n = 44), 3 classes (n = 29) and ≥ 4 antibiotic classes (n = 14) treatment groups. Mean abundance of major genera colour coded and presented as stacked bar graphs present in **c** FMT donors, non-antibiotic AAD (-AAD), 1–2 days, 3–4 days, 5–7 days and ≥ 8 days antibiotic treatment groups, and, **d** FMT donors, non-antibiotic AAD (-AAD), 1 class, 2 classes, 3 classes and ≥ 4 antibiotic classes treatment groups. In panels **a** and **b**, data are presented as mean ± SD. Statistical significance was determined at p < 0.05 and comparisons used Kruskal–Wallis tests with FDR adjusted for multiple comparisons using the Benjamini and Hochberg method. Source data for panels are provided as a Source Data file.

away from FMT donors and pairwise PLS-DA revealed that -Ent AAD vs +Ent AAD metabolomes were significantly different ($R^2Y = 0.564$, $Q^2 = 0.472$ and $p = 2.031 \times 10^{-14}$) (Supplementary Fig. 1B).

Plots of mean SCFA concentrations (acetate, propionate and butyrate) revealed that compared to FMT donors, the -Ent AAD and +Ent AAD faecal metabolomes were significantly depleted in all SCFAs (Supplementary Fig. 1C, E). While there was no

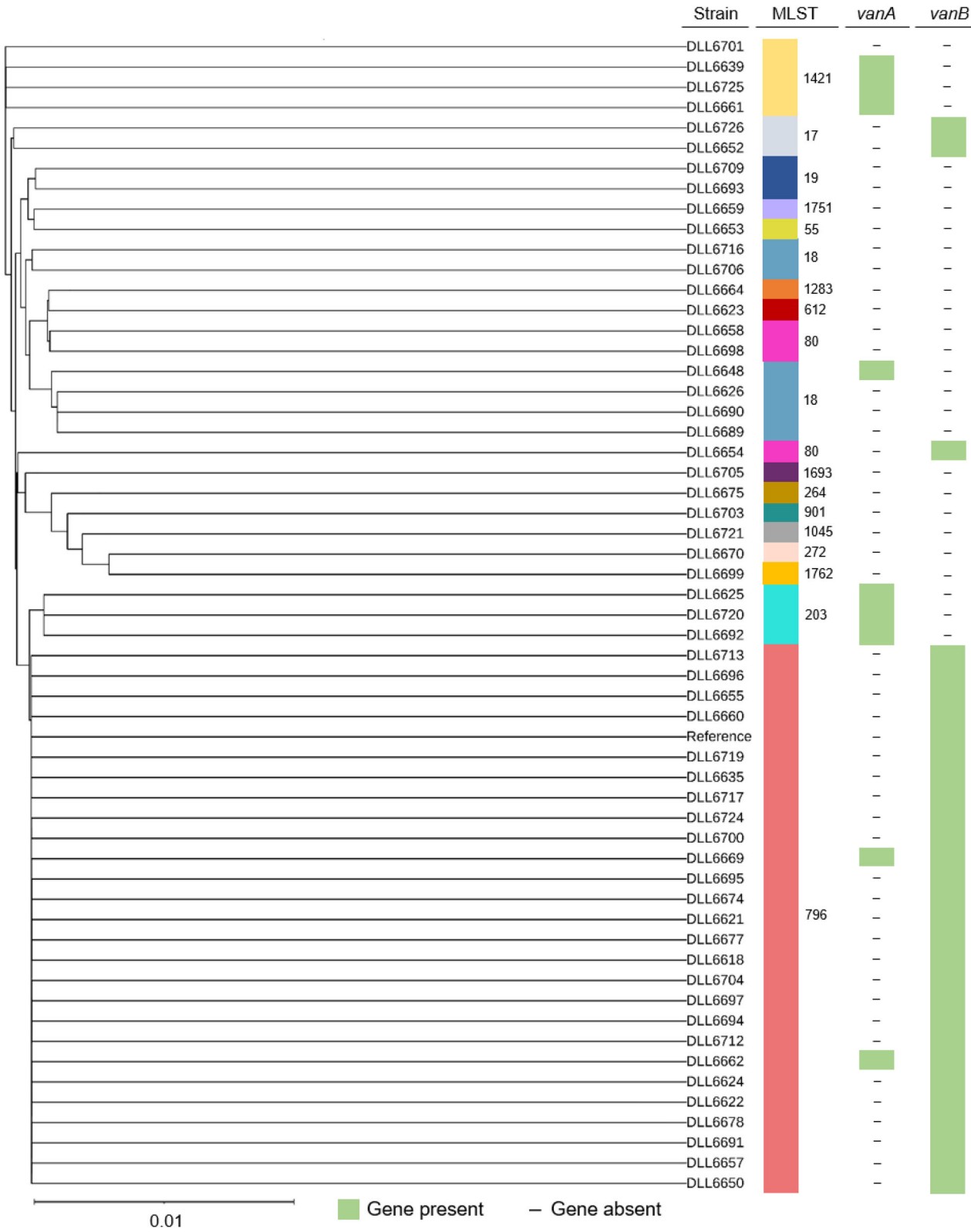

**Fig. 2 | Core phylogenetic analysis, multi-locus sequencing typing (MLST), and vancomycin resistance gene profiling of _E. faecium_ isolates.** Core genome phylogeny, sequence types, and the presence of vancomycin resistance genes _vanA_ and _vanB_ were determined using Nullabor v2.0 pipeline (https://github.com/ tseemann/nullarbor). Analysis was performed against the reference strain, _E. faecium_ Ef_aus00233. In the MLST column, each colour presents a visual representation of sequence type diversity. In the _vanA_ and _vanB_ columns, green denotes gene presence, and – symbol denotes gene absence.

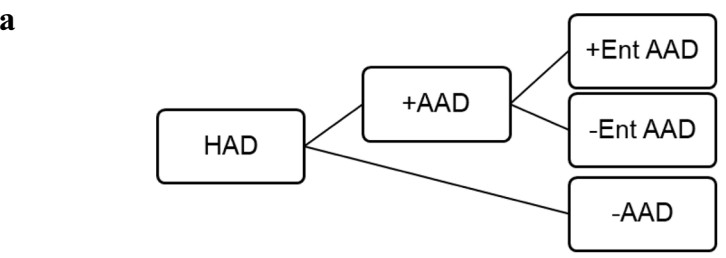

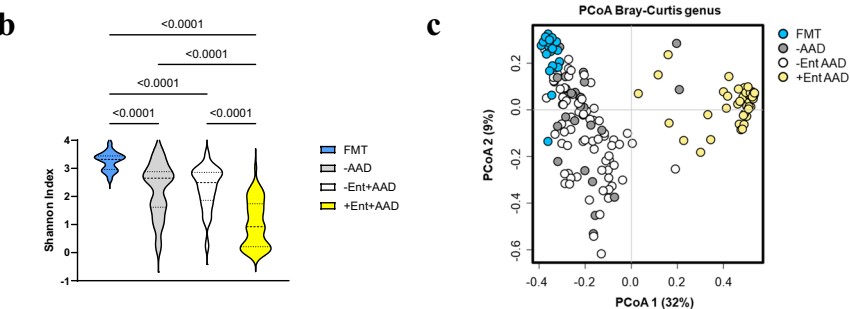

**Fig. 3 | Low diversity AAD with enterococcal proliferation formed a microbially distinct subset of AAD. a** Summary of the HAD and antibiotic-associated diarrhoeal ( + AAD) patient cohorts stratified by enterococcal proliferation. Non-antibiotic AAD (-AAD), AAD without enterococcal proliferation (-Ent+AAD) and AAD with enterococcal proliferation ( + Ent AAD) whose microbiota comprised 25-99% of *Enterococcus* OTUs. **b** Violin plot of Shannon diversity indices assessed species richness and evenness among FMT donors ($n = 20$), -AAD ($n = 30$), -Ent AAD ($n = 76$) and +Ent AAD ($n = 61$) patients. Alpha diversity was estimated from Shannon diversity index (OTU abundances rarefied to 1107 reads). Statistical significance was determined at $p < 0.05$ and comparisons used Kruskal-Wallis tests with FDR adjusted for multiple comparisons using the Benjamini and Hochberg method. Source data provided as a Source Data file. **c** PCoA plot based on the Bray-Curtis dissimilarity assessed microbiota differences of FMT donors ($n = 20$), -AAD ($n = 30$), -Ent AAD ($n = 76$) and +Ent AAD ($n = 61$) patients ($R^2 = 0.328$, $p < 0.001$). Statistical significance was determined at $p < 0.05$ by PERMANOVA. The F statistic two-tailed p-value depicts the significance of the host factor in affecting the community structure, while the PERMANOVA statistic $R^2$ depicts the fraction of variance explained by each factor.

significant difference in faecal acetate between -Ent AAD and +Ent AAD patients, the +Ent AAD metabolome was significantly reduced in propionate (p = 0.037) and butyrate ($p < 0.0001$) concentrations (Supplementary Fig. 1C, E).

### Elevated tyramine/tyrosine ratios as biomarkers of low diversity enterococcal-dominant AAD

Individual metabolites were further assessed for their capacity to distinguish between diarrhoeal groups using the receiver operating characteristics area under the curve (ROC-AUC). Based on the AUC ≥ 0.70 cut-off, several metabolites, including the amino acid L-tyrosine and its derivative desaminotyrosine, differentiated between -Ent AAD and +Ent AAD (Supplementary Table 2).

We noted that while the +Ent AAD metabolome was elevated in several amino acids compared to -Ent AAD, L-tyrosine was the only amino acid significantly depleted in the +Ent AAD metabolome ($p < 0.0001$), with a mean abundance similar to FMT donors (Fig. 4a). Univariate AUC biomarker analysis revealed that reduced L-tyrosine was a possible biomarker of +Ent AAD (AUC = 0.79) (Fig. 4b).

Several bacterial by-products of L-tyrosine metabolism were further analysed, with tyramine of particular interest. Decarboxylation of L-tyrosine into tyramine in the gut is associated with several genera but mainly *Enterococcus*, particularly *E. faecium* and *E. faecalis*[31]. While tyramine was significantly elevated in the +Ent AAD metabolome compared to FMT donors ($p = 0.003$), there was no significant difference between -Ent AAD and +Ent AAD patients (Fig. 4c). Furthermore, univariate AUC biomarker analysis revealed that with an AUC < 0.60, tyramine was a poor biomarker differentiating +Ent AAD from -Ent AAD (Fig. 4d).

The ratio of tyramine to L-tyrosine was calculated for each sample to investigate whether depleted L-tyrosine and elevated tyramine might signify enterococcal utilisation. The tyramine/tyrosine ratio was significantly higher for +Ent AAD patients compared to -Ent AAD patients ($p < 0.0001$) (Fig. 4e), and tyramine/tyrosine ratios performed substantially better in differentiating +Ent AAD with an AUC > 0.80 than tyrosine or tyramine alone (Fig. 4f).

### Metabolite biomarkers of concomitant enterococcal proliferation and CDI

Toxigenic *C. difficile* was detected in -AAD, -Ent AAD and +Ent AAD patients (Supplementary Fig. 2A), however, our analyses showed a lack of genus-level microbiota difference between CDI and non-CDI patients (Supplementary Fig. 2b, d). Despite this, we hypothesised that their metabolomes might present CDI specific-biomarkers. A heatmap of 88 metabolites that best described the variation between CDI and non-CDI patients revealed that -AAD + CDI and -Ent+CDI metabolomes were associated with reduced sugars, sugar alcohols and amino acids compared to their non-CDI counterparts (Fig. 5a). Conversely, the +Ent+CDI and +Ent-CDI metabolomes were similarly enriched in a greater number of metabolites across several compound classes, including alcohols, amines, amino acids, bile acids, and sugars, and reduced in indoles, fatty acids, and phenylpropanoic acids (Fig. 5a).

While the principal component scores plot could not distinguish between CDI and non-CDI groups, multivariate ROC-AUC analysis determined that the PLS-DA classification model showed moderate to high specificity and sensitivity in differentiating each non-CDI and CDI group (AUC > 0.70) (Supplementary Table 3). Furthermore, between-group differences assessed by pairwise PLS-DA analysis found that while +Ent-CDI and +Ent+CDI metabolomes were not significantly different ($p = 1.000$), the difference between -AAD-CDI vs -AAD + CDI approached statistical significance ($p = 0.060$) and was statistically significant between -Ent-CDI and -Ent+CDI ($p = 0.005$) (Supplementary Table 3).

In addition, plots of mean SCFA concentrations (acetate, propionate and butyrate) derived from SCFA profiling revealed that -AAD +

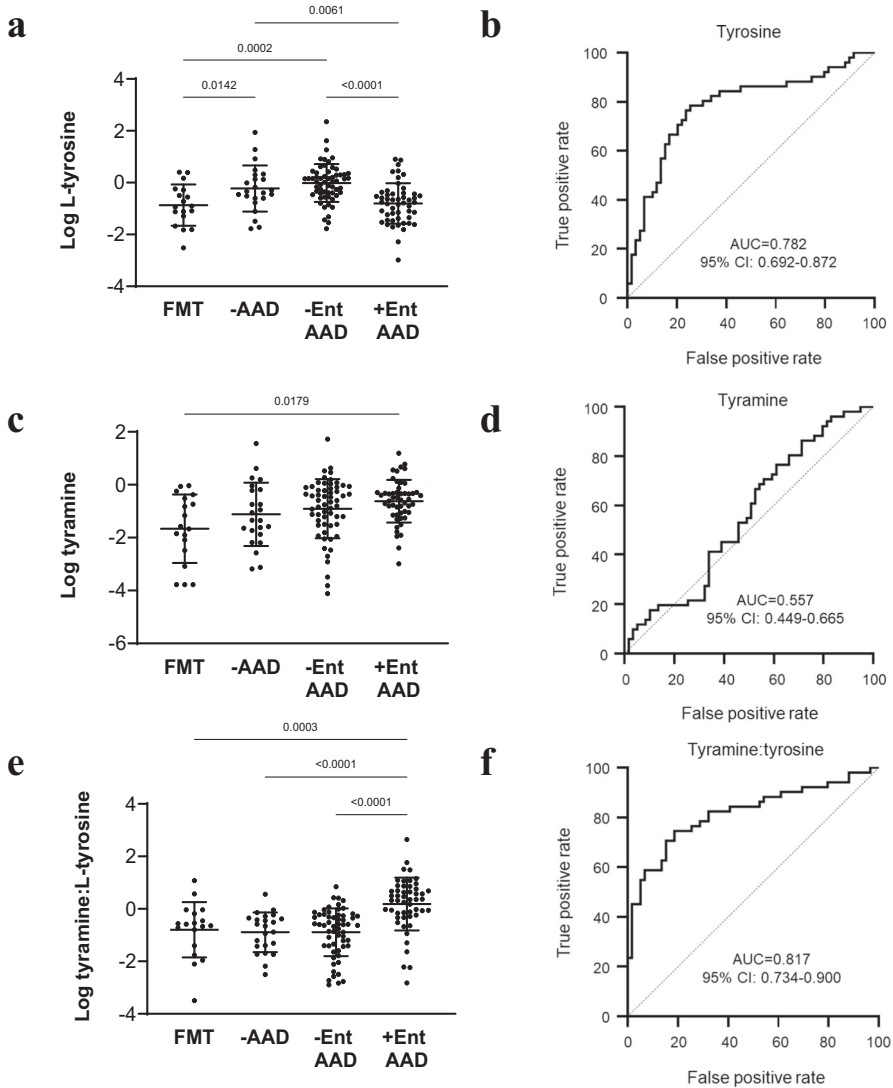

**Fig. 4 | Low diversity enterococcal-dominant AAD is associated with elevated ratios of tyramine to L-tyrosine. a** Dot plot of L-tyrosine abundance. **b** L-tyrosine AUC plot differentiating between -Ent AAD ($n = 59$) from +Ent AAD ($n = 51$) patients. **c** Dot plot of tyramine abundance. **d** Tyramine AUC plot differentiating between −Ent AAD ($n = 59$) from +Ent AAD ($n = 51$) patients. **e** Dot plot of tyramine/tyrosine ratios. **f** Tyramine/tyrosine ratios AUC plot differentiating between -Ent AAD ($n = 59$) from +Ent AAD ($n = 51$) patients. Data presented as mean ± SD in panels **a, c** and **e** for FMT donors ($n = 20$), -AAD ($n = 23$), -Ent AAD ($n = 59$) and +Ent AAD ($n = 51$) patients. Statistical significance was determined at $p < 0.05$ and comparisons used Kruskal-Wallis tests with FDR adjusted for multiple comparisons using the Benjamini and Hochberg method. Source data is provided as a Source Data file.

CDI and -Ent+CDI patients were elevated in acetate and butyrate, with mean concentrations similar to FMT donors (Fig. 5c, d). In contrast, -Ent-CDI and +Ent-CDI patients were significantly reduced in acetate compared to FMT donors, but the only significant between-group difference observed was between -Ent+CDI and +Ent-CDI patients ($p = 0.016$) (Fig. 5d). -AAD-CDI, -Ent-CDI, +Ent+CDI and +Ent+CDI patients were significantly depleted in butyrate. Similarly, the only significant between-group difference observed was between -Ent+CDI and +Ent-CDI patients ($p = 0.013$) (Fig. 5d). Univariate AUC biomarker analysis determined that acetate and butyrate, with AUC values > 0.70, were important biomarkers that could differentiate -Ent+CDI patients from -Ent-CDI patients (Supplementary Fig. 3a, b).

**Proline Stickland fermentation by-products as biomarkers of concomitant enterococcal proliferation and CDI**

Using ROC-AUC, individual metabolites were further assessed for their capacity to distinguish between CDI and non-CDI groups. Supplementary Tables 4-6 detail the metabolites that distinguished between -AAD-CDI and -AAD + CDI, -Ent-CDI and -Ent+CDI, and +Ent-CDI and +Ent+CDI. We detected several Stickland by-products, including 5-aminovaleric acid (from L-proline), 4-methylvaleric acid (4-MPA) (from L-leucine), isovalerate (from L-leucine), isobutyrate (from L-valine) and desaminotyrosine (from L-tyrosine) (Supplementary Tables 4−6). Non-enterococcal CDI patients were also associated with elevated indole/tryptophan ratios (Supplementary materials and Supplementary Fig. 4).

5-aminovaleric acid was elevated in -Ent+CDI and +Ent+CDI patients compared to FMT donors and their non-CDI counterparts, but between-group differences were not significant (Fig. 6a). However, univariate AUC biomarker analysis determined that 5-aminovaleric acid was a potential biomarker differentiating +Ent+CDI from +Ent-CDI patients (AUC = 0.735) (Fig. 6b). We calculated the ratio of 5-aminovaleric acid to proline for each individual to investigate whether together, depleted proline and elevated 5-aminovaleric acid might signify *C. difficile* utilisation. Mean 5-aminovaleric acid/proline ratios were reduced in non-CDI patients compared to their CDI counterparts but the differences in 5-aminovaleric acid/proline ratios between CDI groups and their non-CDI counterparts were not statistically significant

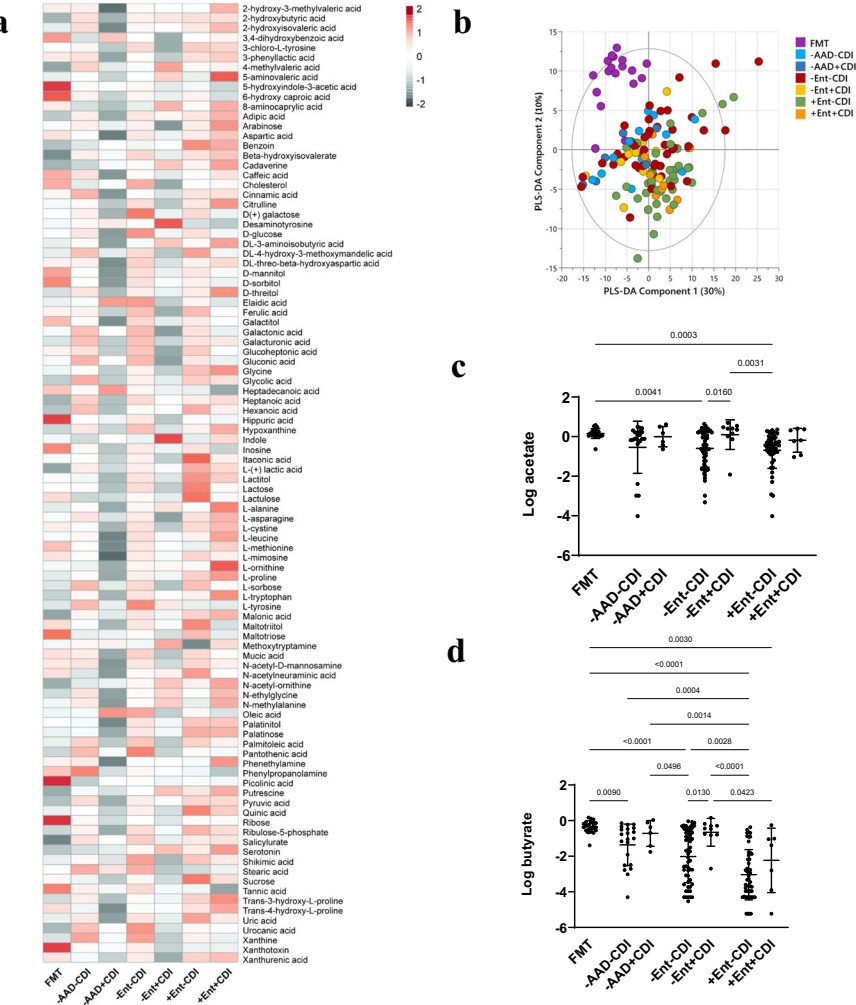

**Fig. 5 | Non-antibiotic and non-enterococcal CDI metabolomes shared a reduction in sugars and amino acids compared to enterococcal CDI and non-CDI metabolomes. a** Heatmap of metabolite abundances detected by untargeted GC-MS profiling that differentiated FMT donors ($n = 18$), -AAD-CDI ($n = 15$), -AAD + CDI ($n = 7$), -Ent-CDI ($n = 48$), -Ent+CDI ($n = 11$), +Ent-CDI ($n = 42$) and +Ent+CDI ($n = 9$) patients. All metabolites were normalised, Pareto scaled, and log-transformed. Metabolites with VIP scores > 1.0 and p(corr) values > 0.5 and < −0.5 were identified as a subset of metabolites with the highest potential as biomarkers. For detailed VIP and p(corr) values, see Source Data file. Each cell corresponded to the mean abundance for each metabolite per group. Dark grey indicated the lowest and red the highest value. **b** PLS-DA scores plot for FMT donors (purple), -AAD-CDI (light blue), -AAD + CDI (dark blue), -Ent-CDI (red), -Ent+CDI (yellow), +Ent-CDI

(green) and +Ent+CDI (orange) patients. Each point represented an individual specimen. Model cross-validation ($R^2Y = 0.244$, $Q^2 = 0.057$, $p = 0.080$ CV-ANOVA). See Source Data file for all model details. **c** Dot plot of acetate concentrations (μg per mg of fresh weight specimen (FW)). **d** Dot plot of butyrate concentrations (μg per mg of fresh weight specimen (FW)). SCFAs GC-MS profiling data are presented as mean ± SD in panels **c** and **d** for FMT donors ($n = 20$), -AAD-CDI ($n = 21$), -AAD + CDI ($n = 6$), -Ent-CDI ($n = 56$), -Ent+CDI ($n = 10$), +Ent-CDI ($n = 49$) and +Ent+CDI ($n = 7$) patients. In panels c and d, statistical significance was determined at $p < 0.05$ and comparisons used Kruskal-Wallis tests with FDR adjusted for multiple comparisons using the Benjamini and Hochberg method. Source data provided as a Source Data file.

(Fig. 6c). However, univariate AUC biomarker analysis determined that 5-aminovaleric acid/proline ratios performed similarly as a biomarker of -Ent+CDI (AUC = 0.718) (Fig. 6d) as 5-aminovaleric acid alone.

## L-leucine and L-valine Stickland fermentation by-products as biomarkers of CDI without enterococcal proliferation

Compared to FMT donors, 4-MPA was elevated with antibiotic usage and enterococcal dominance and significantly elevated in -AAD-CDI (p = 0.040) and -Ent+CDI patients (p = 0.040) but between-group differences for -AAD-CDI vs -AAD + CDI and -Ent-CDI vs -Ent+CDI patients were not significant (Fig. 7a). However, univariate ROC-AUC determined that 4-MPA approached significance as a biomarker differentiating -Ent+CDI from -Ent-CDI patients (AUC = 0.682) (Fig. 7b). We calculated the ratio of L-leucine and 4-MPA for each individual to investigate whether together, depleted L-leucine and elevated 4-MPA might signify *C. difficile* utilisation. Compared to FMT donors, 4-MPA/

L-leucine ratios were elevated in all groups, except -Ent+CDI patients who shared similarly reduced mean 4-MPA/L-leucine ratios as FMT donors (Fig. 7c). The mean 4-MPA/L-leucine ratio was significantly elevated in -Ent+CDI compared to FMT donors (p = 0.012) and -Ent-CDI patients (p = 0.036) (Fig. 7c). Univariate AUC analysis determined that 4-MPA/L-leucine ratios performed substantially better in differentiating -Ent+CDI from -Ent-CDI (AUC > 0.800) (Fig. 7d), than 4-MPA alone.

In a separate analysis, SCFA profiling via GC-MS detected L-leucine and L-valine Stickland fermentation products, isovalerate and isobutyrate. Compared to FMT donors, the isovalerate and isobutyrate mean concentration decreased among non-CDI patients with antibiotic treatment and enterococcal dominance but were elevated in -AAD + CDI and -Ent+CDI patients, similar to FMT donors (Fig. 7e, f). However, analysis of CDI and non-CDI between-group differences determined only -Ent-CDI compared to -Ent+CDI patients were significantly reduced in isovalerate and isobutyrate (p = 0.002 and

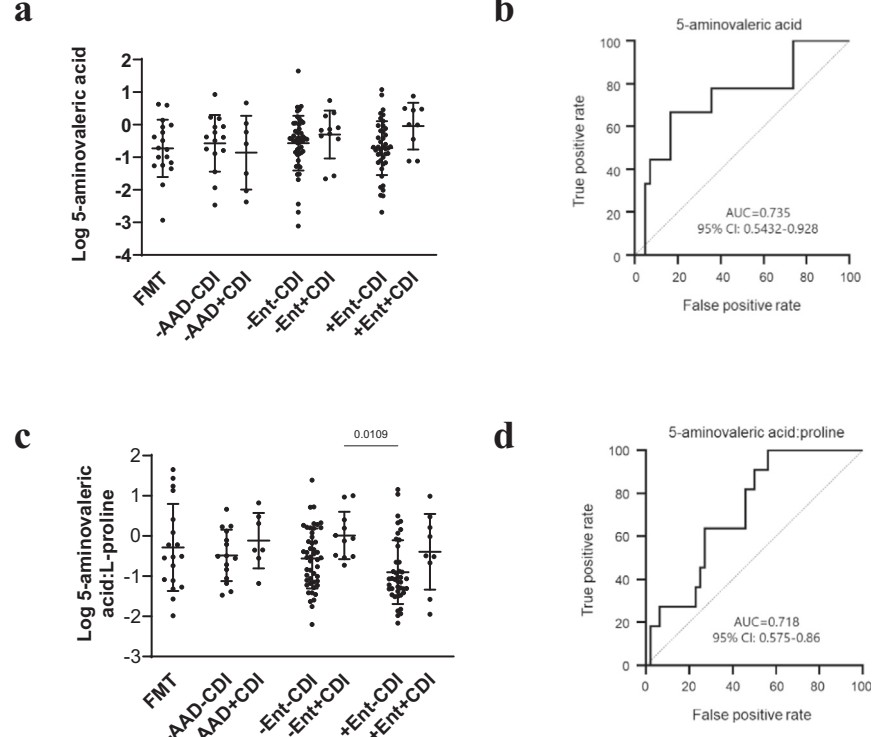

**Fig. 6 | Enterococcal CDI was associated by-products of L-proline Stickland fermentation. a** Dot plot of 5-aminovaleric acid abundance as detected by untargeted GC-MS profiling. **b** 5-aminovaleric acid AUC plot differentiating +Ent-CDI ($n$ = 42) from +Ent+CDI ($n$ = 9) patients. **c** Dot plot of 5-aminovaleric/L-proline ratios. **d** 5-aminovaleric acid/L-proline ratios AUC plot differentiating +Ent-CDI ($n$ = 42) from +Ent+CDI ($n$ = 9) patients. Data presented as mean ± SD in panels

**a** and **c** for FMT donors ($n$ = 18), -AAD-CDI ($n$ = 15), -AAD + CDI ($n$ = 7), -Ent-CDI ($n$ = 48), -Ent+CDI ($n$ = 11), +Ent-CDI ($n$ = 42) and +Ent+CDI ($n$ = 9) patients. In panels **a** and **c**, statistical significance was determined at $p < 0.05$ and comparisons used Kruskal-Wallis tests with FDR adjusted for multiple comparisons using the Benjamini and Hochberg method. Source data provided as a Source Data file.

$p = 0.0004$, respectively) (Fig. 7e, f). Univariate ROC-AUC analysis determined isovalerate (AUC = 0.830) and isobutyrate (AUC = 0.886) as significant biomarkers differentiating -Ent+CDI patients from -Ent-CDI patients (Fig. 7g, h).

### L-tyrosine Stickland fermentation by-product as biomarkers of CDI without enterococcal proliferation

Desaminotyrosine was reduced in all diarrhoeal groups compared to FMT donors except for -Ent+CDI patients with a mean abundance that exceeded that of FMT donors (p = 0.014) and their non-CDI counterparts (p = 0.129) (Fig. 8a). Univariate AUC biomarker analysis determined that desaminotyrosine was a significant biomarker differentiating -Ent+CDI patients from -Ent-CDI patients (AUC = 0.720) (Fig. 8b). We calculated the ratio of L-tyrosine and desaminotyrosine for each individual to investigate whether together, depleted L-tyrosine and elevated desaminotyrosine might signify *C. difficile* utilisation. Compared to FMT donors, desaminotyrosine/L-tyrosine ratios were reduced in all diarrhoeal groups, except in -AAD + CDI and -Ent+CDI groups who shared a similarly elevated mean desaminotyrosine/L-tyrosine ratio as FMT donors (Fig. 8c). The difference in 5- desaminotyrosine/L-tyrosine ratios between CDI groups and their non-CDI counterparts was statistically significant between -Ent-CDI and -Ent+CDI groups (p = 0.025) (Fig. 8c). Univariate AUC biomarker analysis found desaminotyrosine/tyrosine ratios performed better in differentiating Ent+CDI patients (AUC = 0.807) than desaminotyrosine on its own (Fig. 8d).

### Discussion

This study characterised the faecal microbiota and metabolomes of HAD patients with respect to antibiotic treatment before specimen collection and CDI status. 16 S rRNA gene sequencing combined with untargeted and SCFA GC-MS-based profiling approaches revealed that HAD, AAD and CDI patients were associated with diverse faecal microbiota and metabolome compositions. The study found that AAD patients with high proportions of *Enterococcus* (predominantly vancomycin-resistant *E. faecium*) were associated with extended antibiotic exposure and combination antibiotic treatment and could be differentiated by elevated biomarkers of tyrosine decarboxylation. Furthermore, controlling for antibiotic usage, CDI microbiota did not differ significantly from non-CDI but could be metabolomically distinguished from non-CDI by biomarkers of Stickland fermentation and colonisation resistance shared with FMT donors.

As a common healthcare-associated pathogen, dense gastrointestinal colonisation of VRE can lead to systemic infection and transmission, putting vulnerable patients at risk. In this study, the prevalence and high abundance of *Enterococcus* detected in the AAD and CDI microbiota strongly correlated with extended antibiotic treatment and combination antibiotic therapy. By sampling HAD patients hospital-wide rather than in a single hospital ward, we found that enterococcal-dominant AAD formed a distinct subset of AAD. *Enterococcus* culturing revealed that the dominant species in these specimens was *E. faecium*; over half of these isolates encoded vancomycin resistance determinants *vanA* or *vanB*, and approximately 46% belonged to the epidemic *vanB* ST796 clonal group. The ST796 clonal group, first identified in 2012 in a Melbourne hospital[32], was responsible for 53% of all *E. faecium* bacteraemia cases in Melbourne hospitals by 2015[33]. It is unknown whether patients in this study were carrying ST796 *E. faecium* before hospital admission or were colonised following healthcare-associated transmission, as faecal samples were only collected after developing diarrhoea following two or more days in hospital.

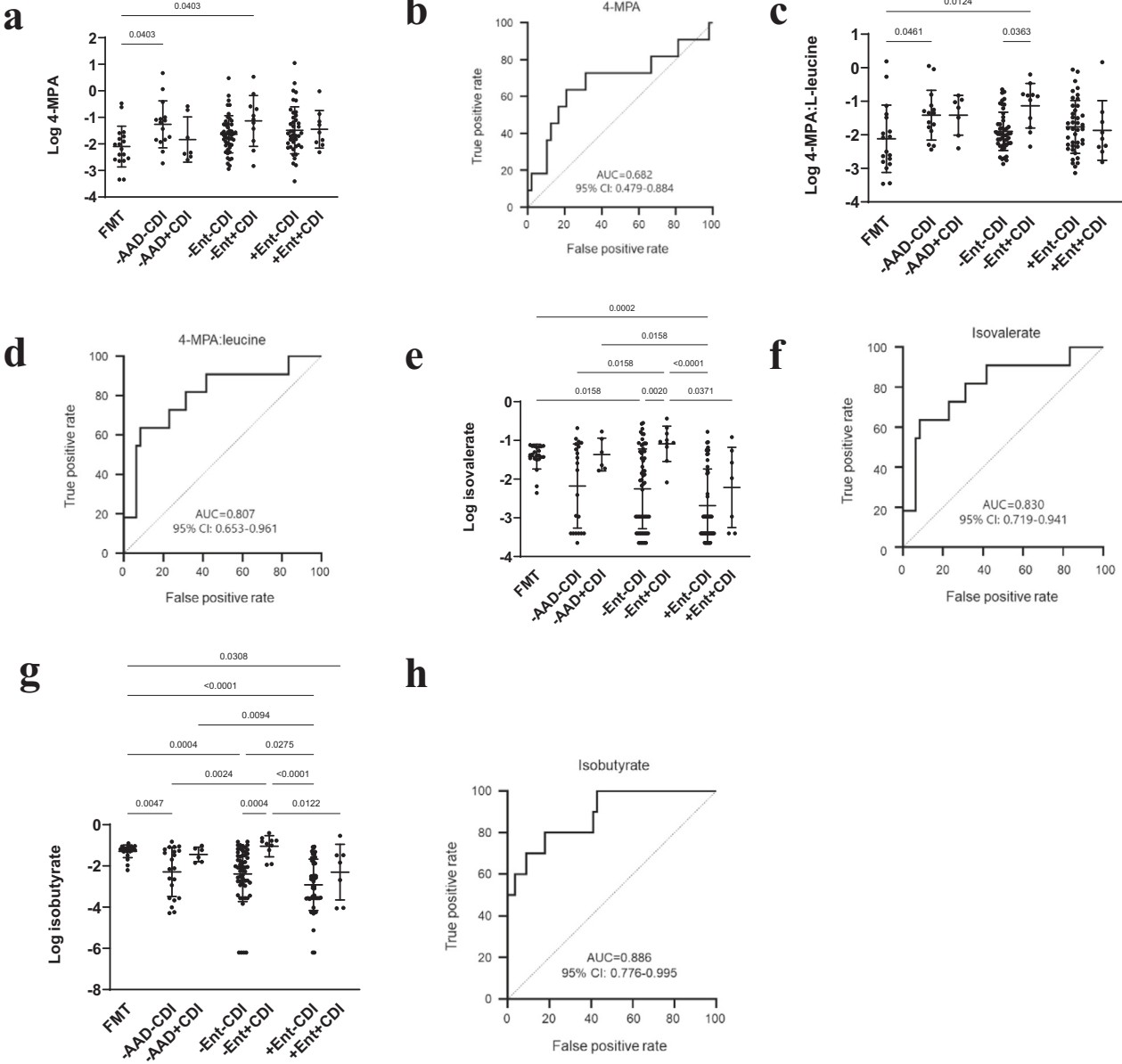

**Fig. 7 | Non-enterococcal CDI was associated with by-products of L-leucine and L-valine Stickland fermentation. a** Dot plot of 4-MPA abundance as detected by untargeted GC-MS profiling. **b** AUC plot of 4-MPA differentiating +Ent-CDI ($n = 42$) from +Ent+CDI ($n = 9$) patients. **c** Dot plot of 4-MPA/L-leucine ratios. **d** AUC plot of 4-MPA/L-leucine ratios differentiating +Ent-CDI ($n = 42$) from +Ent+CDI ($n = 9$) patients. **e** Dot plot of isovalerate concentrations as detected by SCFA GC-MS profiling. **f** Isovalerate AUC plot differentiating +Ent-CDI ($n = 49$) from +Ent+CDI ($n = 7$). **g** Dot plot of isobutyrate concentrations as detected by SCFA GC-MS profiling. **h** Isobutyrate AUC plot differentiating +Ent-CDI ($n = 49$) from +Ent+CDI

($n = 7$). Untargeted GC-MS profiling data is presented as mean ± SD in panels **a** and **c** for FMT donors ($n = 18$), -AAD-CDI ($n = 15$), -AAD + CDI ($n = 7$), -Ent-CDI ($n = 48$), -Ent+CDI ($n = 11$), +Ent-CDI ($n = 42$) and +Ent+CDI ($n = 9$) patients. SCFA GC-MS profiling data is presented as mean ± SD in panels **e** and **g** for FMT donors ($n = 20$), -AAD-CDI ($n = 21$), -AAD + CDI ($n = 6$), -Ent-CDI ($n = 56$), -Ent+CDI ($n = 10$), +Ent-CDI ($n = 49$) and +Ent+CDI ($n = 7$) patients. In panels **a, c, e** and **g**, statistical significance was determined at $p < 0.05$ and comparisons used Kruskal-Wallis tests with FDR adjusted for multiple comparisons using the Benjamini and Hochberg method. Source data provided as a Source Data file.

*Enterococcus*, particularly *E. faecium* and *E. faecalis*, have been shown to produce large amounts of tyramine in vitro[31]. In this study, biomarker assessment revealed that elevated ratios of tyramine to tyrosine were a potential indicator of enterococcal proliferation. However, depleted tyrosine and elevated tyramine correlating with elevated *Enterococcus* have only been observed in vivo in mice with graft-versus-host-disease[34] and clinically in Parkinson's disease patients treated with L-DOPA (dopamine)[35]. *Enterococcus* has been shown to decarboxylate tyrosine and dopamine into tyramine at a similar rate in vitro[35], however, in the current study it is unknown whether unaccounted L-DOPA treatment was a factor in enterococcal proliferation and elevated tyramine but warrants further investigation.

CDI has been associated with significantly perturbed microbiota among critically ill patients with high antibiotic usage. We found that CDI was associated with a spectrum of microbiota dysbiosis that reflected contrasting antibiotic treatment rather than CDI status. Stratification of CDI patients with respect to antibiotic treatment and enterococcal dominance also determined that microbiota differences between CDI and non-CDI patients were insignificant, supporting in vivo mouse CDI susceptibility studies that have shown that *C. difficile* could colonise and cause disease in a spectrum of antibiotic perturbed gut environments[13,18,19].

While the CDI and non-CDI microbiota did not differ significantly, non-antibiotic and non-enterococcal dominant CDI patients were

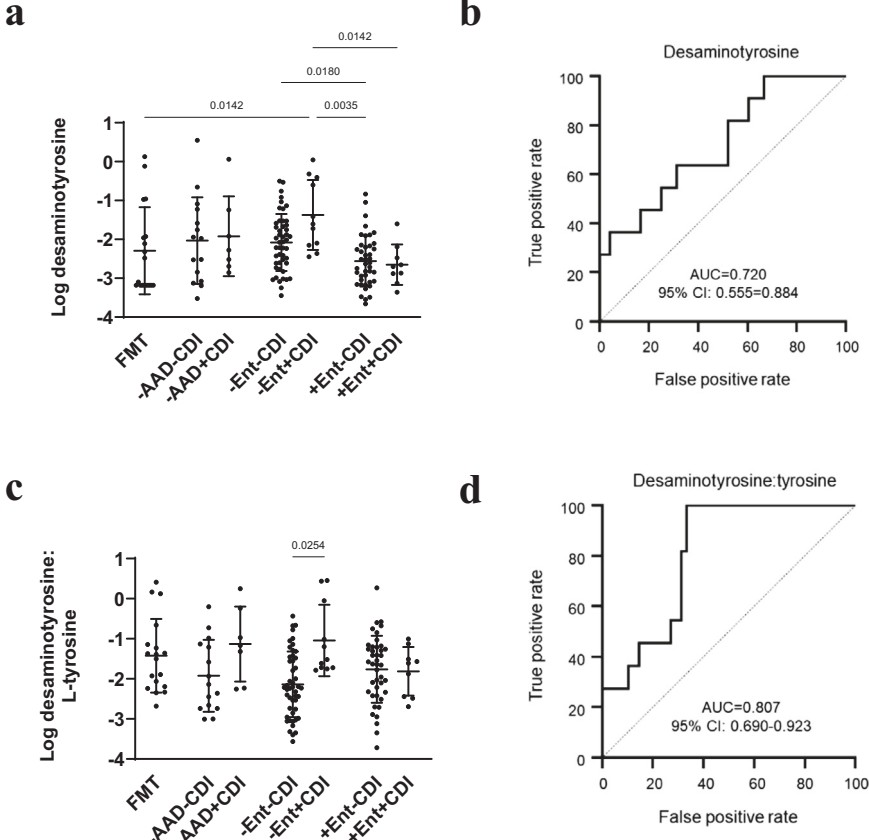

**Fig. 8 | Non-enterococcal CDI was associated with by-products of tyrosine Stickland fermentation. a** Dot plot of desaminotyrosine abundance.
**b** Desaminotyrosine AUC plot differentiating -+Ent-CDI (*n* = 42) from +Ent+CDI (*n* = 9) patients. **c** Dot plot of desaminotyrosine/L-tyrosine ratios.
**d** Desaminotyrosine/L-tyrosine ratios AUC plot differentiating +Ent-CDI (*n* = 42) from +Ent+CDI (*n* = 9) patients. Data presented as mean ± SD in panels **a** and **c** for

FMT donors (*n* = 18), -AAD-CDI (*n* = 15), -AAD + CDI (*n* = 7), -Ent-CDI (*n* = 48), -Ent +CDI (*n* = 11), +Ent-CDI (*n* = 42) and +Ent+CDI (*n* = 9) patients. Statistical significance was determined at *p* < 0.05 and comparisons used Kruskal-Wallis tests with FDR adjusted for multiple comparisons using the Benjamini and Hochberg method. Source data provided as a Source Data file.

found to have markedly different metabolome profiles to their non-CDI counterparts. Importantly, they shared metabolite features with FMT donors with reduced amino acids, sugars and elevated SCFAs. These differences were unexpected, given that there was no detectable microbiota variation between CDI patients and their non-CDI counterparts. Metabolite similarities with FMT donors suggested that the non-antibiotic and non-enterococcal CDI metabolomes were healthier, possibly due to unaccounted commensal bacteria present at levels below the detection limit of the 16 S rRNA-based microbiota analysis (i.e., less than 0.01% of microbiota). This may, in part, reflect diversity in antibiotic treatment, as CDI patients were weakly associated with higher cephalosporin usage. Cephalosporin usage has been associated with proteobacteria blooms; these were not detected in the current study. While CDI and non-CDI patients were clinically similar, the CDI cohort may reflect individuals whose starting microbiota was less perturbed at hospital admission.

In the current study, several *C. difficile*-specific biomarkers derived from Stickland fermentation of branched-chain amino acids could differentiate CDI patients from non-CDI patients. These results suggest that in human CDI, *C. difficile* may preferentially catabolise amino acids over sugars during infection, which is consistent with the hypothesis that the availability of faecal amino acids enhances CDI susceptibility[27,36]. Specifically, 4-MPA, isobutyrate, isovalerate, 5-aminovaleric acid and desaminotyrosine formed putative CDI biomarkers. However, the contributions of these Stickland products may be due to unaccounted-for taxa as the presence of other Stickland fermenting clostridia, including *Clostridium bifermentans, Clostridium*

*botulinum, Clostridium sporogenes, Paraclostridium sordellii* and *Peptostreptococcus anaerobius*, were not detected in the current study[37]. Similarly, while *C. difficile* can produce small amounts of desaminotyrosine during its growth phase, desaminotyrosine has not been previously associated with CDI[27]. Instead, elevated desaminotyrosine has been associated with microbiota response to infection and inflammation by commensal Stickland fermenting bacteria such as *C. orbiscindens*, which were not detected in the current study[38,39]. However, the detection of desaminotyrosine and other Stickland products may also reflect *C. difficile* metabolic diversity in response to microbiota competition for the amino acids.

Due to the elevated abundance of isovalerate and 4-MPA, L-leucine may be a more energetically favourable *C. difficile* Stickland substrate in microbially competitive environments[27]. The reduction in amino acids in vivo signals toxin production over vegetative cell growth and, therefore, elevated 4-MPA in the amino acid-depleted non-enterococcal dominant CDI metabolomes might be consistent with toxin production[30]. While we did not directly detect *C. difficile* toxins as part of the study, the correlation between 4-MPA and toxin production needs to be explored further to determine whether elevated 4-MPA is due to *C. difficile* or other clostridia that compete with *C. difficile*.

CDI in vivo mouse studies have shown that *C. difficile* metabolism changes during CDI progression, while other in vivo CDI mouse studies have shown that *C. difficile* can induce microbiota response. The metabolite differences observed in non-antibiotic and non-enterococcal dominant CDI patients compared to their non-CDI counterparts may

also reflect a concomitant microbiota and *C. difficile* response during early disease, toxin production and inflammation. Non-enterococcal dominant CDI patients were associated with elevated ratios of indole to tryptophan, similar to FMT donors, but indole is not produced by *C. difficile*. Instead, indole is one of several bacterial metabolites produced from the fermentation of dietary tryptophan with putative anti-inflammatory and interkingdom signalling roles[40,41]. While elevated indole may suggest the presence of undetected indole producers, in vitro studies have shown that *C. difficile* can induce other bacteria to overexpress indole to remove other indole-sensitive commensals[42].

The role of SCFAs in CDI is unclear. Elevated butyrate and acetate detected in non-antibiotic and non-enterococcal dominant CDI metabolomes might indicate contributions by undetected commensal SCFA-producing bacteria. At the same time, elevated butyrate and acetate may also be an upregulated response by commensals in response to infection as butyrate can down-regulate and attenuate intestinal inflammation, and high concentrations of SCFAs are inhibitory to bacteria[23]. However, butyrate accumulation may also indicate impaired uptake by IECs across the apical membrane due to toxin-mediated damage and inflammation[43]. Lastly, elevated acetate and butyrate might also reflect *C. difficile* contributions in response to amino acid competition. In vitro studies have shown that not only can *C. difficile* produce butyrate in the absence of amino acid Stickland substrates, but butyrate export coincided with toxin secretion during the late phases of *C. difficile* growth[36].

While little is known about the relationships between concomitant CDI and VRE proliferation, epidemiologically, these patients are associated with poorer outcomes[7,8]. A recent in vitro transcriptomic study found that *E. faecalis* could reshape the metabolome by depleting ornithine and introducing fermentable amino acids (notably arginine), during which *C. difficile* altered its metabolism in favour of Stickland fermentation with predicted end-products of isoleucine and proline metabolism[17]. In the current study, while *E. faecium*-dominant CDI and non-CDI metabolomes were characterised by elevated amino acids, L-tyrosine was depleted among enterococcal-dominant patients. Only the proline Stickland by-product, 5-aminovaleric acid, differentiated between the two, supporting previous observations that proline is the preferred *C. difficile* energy source[18,27,36]. At the same time, biomarkers of tyrosine decarboxylation in enterococcal-dominant AAD were significant, suggesting concomitant enterococcal proliferation and CDI might capitalise on two different amino acid pathways to colonise, persist and possibly inhibit other commensals. Together, these biomarkers of enterococcal dominance and CDI provide an insight into the nutritionally segregated nature of *Enterococcus* and *C. difficile* colonisation.

Detecting *C. difficile* biomarkers in clinical CDI for diagnostic purposes is difficult due to *C. difficile* metabolic flexibility and the heterogeneous nature of nutrient availability in the gut environment. While microbiota and metabolomic studies are not feasible as part of routine microbiological diagnostics, established screening practices for the detection of toxigenic *C. difficile*, coupled with putative *C. difficile* and enterococcal biomarkers, may better predict CDI susceptibility and guide treatment. With metabolite features similar to FMT donors, CDI patients associated with low antibiotic usage might retain sufficient microbiota to compete for amino acids and decolonise *C. difficile* upon cessation of antibiotic treatment (if appropriate). Conversely, patients with enterococcal-dominant CDI microbiota might benefit from alternative supportive care, including FMT or microbiota drugs that have been efficacious in treating recurrent CDI and decolonising concomitant VRE and *C. difficile*[44]. Furthermore, the metabolome profiles presented in this study suggest that the amino acids proline and tyrosine are significant substrates in CDI and VRE proliferation. Therefore, their modification via dietary interventions might also be of some therapeutic benefit.

The current study had several limitations that could be addressed with longitudinal and prospective studies with a larger cohort of CDI patients. CDI patients formed a smaller cohort that may have reduced the statistical significance of biomarker assessment. Pre-admission antibiotic treatment, unaccounted medications, comorbidities and dietary interventions were potential contributors to the microbiota and metabolome heterogeneity. Non-diarrhoeal hospitalised controls are also required to assess the large-scale shifts in microbiota and metabolite composition associated with diarrhoea and medical interventions such as bowel washouts[45]. Lastly, butyrate producers and other Stickland bacteria with overlapping niches may be missed due to the limitations associated with amplicon sequencing compared to metagenome sequencing, and potentially over-aggressive filtering of OTUs may result in the loss of rarer taxa.

In summary, this study showed that CDI was associated with different metabolite biomarkers that correlated with increasing antibiotic-associated dysbiosis and proliferation of opportunistic bacteria such as *Enterococcus*. CDI microbiota reflected the effects of contrasting antibiotic exposure rather than *C. difficile* toxin-mediated clearance of commensal microbes. Metabolite biomarkers suggest a dynamic relationship between *C. difficile* and the resident microbiota, with *C. difficile* adopting different strategies in response to changing gastrointestinal conditions and microbiota resistance to infection. The particular makeup of metabolite biomarkers suggests increased colonisation resistance by resident microbiota in response to early *C. difficile* establishment. Taken together, this study provides a unique insight into the structure of the CDI gut microbiota and metabolome with increasing dysbiosis that provides the basis for further study into *C. difficile* metabolism and pathogenesis.

## Methods

### Study approval
Approval for the use of FMT specimens was obtained from Bellberry Human Research Ethics Committee (HREC 2020-03-288). Monash University Human Research Ethics Committee (HREC 29548) also approved the use of these specimens. Approval for the use of HAD specimens was obtained from Monash Health Human Research Ethics Committee (HREC 49004) and Monash University Human Research Ethics Committee (HREC 28455). Donor, patient and specimen evaluations for these cohorts are found in Supplementary Materials. All participants consented to this study.

### Specimen collection
Diarrhoeal specimens fulfilling the inclusion criteria (See Supplementary Materials) were selected from diarrhoeal specimens submitted for *C. difficile* testing at Monash Health, a 640-bed teaching and research hospital in Victoria, Australia. *C. difficile* was the only gastrointestinal pathogen recorded as part of the study. Monash Health Microbiology Laboratory testing protocols included detecting the *C. difficile* glutamate dehydrogenase (GDH) enzyme (LIAISON XL, DiaSorin, Sallugia, Italy). GDH-positive samples were subsequently tested for toxin B (*tcdB*) and binary toxin (*cdtA*) genes (GeneXpert, Cepheid, Sunnyvale, California, USA). A positive PCR result represented clinically significant CDI. Twenty faecal microbiota transplant (FMT) donor specimens donated to the BiomeBank, an FMT clinic in Thebarton, South Australia, were examined alongside HAD patients. The health status and eligibility of all donors were screened following BiomeBank's screening protocols that included an interview, medical assessment, blood, and stool screening (See Supplementary Methods). All specimens following collection were separated into triplicate samples and stored at −80 °C until DNA isolation and metabolite extractions were performed.

### 16 S rRNA amplicon sequencing and microbiota analysis
Total DNA was isolated from 150 mg of faecal specimens using the Bioline ISOLATE II Fecal DNA Kit (Bioline, Eveleigh, Australia).

PCR amplification of the V3 and V4 variable regions of the 16 S rRNA gene was performed using the forward primer 338 F 5'-ACTCCTACGGGAGGCAGCAG-3' and the reverse primer 806 R 5'-GGACTACHVGGGTWTCTAAT-3'. As previously described, the primers also contained barcodes, spacers, and Illumina sequencing linkers[46]. Sequencing was performed on the Illumina MiSeq platform using 2 ×300 bp paired-end sequencing. The 16 S rRNA sequence data are available from the NIH Sequence Read Archive (SRA) under BioProject PRJNA986597, accession numbers SRR24999020, SRR24999021, SRR24999022 and SRR249990213.

Paired-end Illumina sequences were compiled using the Fastq-Join algorithm and taxonomic assignments performed in QIIME v.1.9.1 against the GreenGenes database and QIIME default parameters[47]. Bacterial sequences were clustered into operational taxonomic units (OTUs) at a 99% identity threshold using the Uclust algorithm[48]. OTUs that comprised less than 0.01% of the total microbiota were removed, leaving 5925 OTUs.

Using OTU abundances, alpha diversity was determined by calculating the Shannon diversity index and statistical significance was determined using the non-parametric Mann-Whitney U and Kruskal-Wallis H tests. Beta diversity calculations from OTU abundances were determined by generating a genus-level Bray-Curtis dissimilarity matrix and visualised using principal component analysis plots (PCoA). Microbial community differences were investigated using permutational multivariate analysis of variances (PERMANOVA) and statistical significance was determined at $p < 0.05$. Alpha and beta diversity was calculated and visualised in Calypso[49]. Other than alpha diversity measures that used rarefed data, all statistical analysis was carried out using the OTU table that was square root transformed and Total Sum Scaling (TSS) normalised.

The taxonomic composition of HAD and FMT microbiota was assessed by determining the mean abundance of OTUs at a phylum, family, and genus level. Differential abundance analysis was conducted using Analysis of Compositions of Microbiomes (ANCOM)[50]. Putative taxonomic biomarkers were assessed using the taxa with AUC values greater than 0.70 were retained. Taxonomic comparisons were calculated and visualised in Calypso[49].

### Culturing of enterococcal dominant specimens
All clinical samples determined to be enterococcal-dominant via 16 S rRNA sequencing were plated onto Horse blood agar (HBA; Blood Agar Base (Oxoid) with 5% horse blood) and Slanetz and Bartley agar (Oxoid), followed by incubation at 37 °C for 24-48 hours. Colonies were sub-cultured onto HBA to yield pure cultures, identification confirmed using MALDI-TOF mass spectroscopy[51], and DNA purified using the DNeasy Blood and Tissue kit, with DNA sequencing performed using Illumina MiSeq v2 to achieve paired end 150 bp reads. De novo genome assemblies were prepared using SPAdes genome assembler and annotated using Prokka. Core genome phylogeny, sequence types, and the presence of vancomycin resistance genes *vanA* and *vanB* were determined using Nullabor v2.0 pipeline (https://github.com/tseemann/nullarbor), with analysis performed in comparison to the reference strain, E. *faecium* Ef_aus00233[33]. The core phylogenetic tree was visualised using Interactive Tree Of Life (iTOL) v5[52]. The *E. faecium* sequence data is available from the NCBI database under BioProject ID PRJNA1015000, accession numbers SAMN37345311- SAMN37345366).

### Untargeted metabolomic profiling (gas chromatography-mass spectrometry)
Faecal metabolites were extracted and derivatised, followed by analysis on Agilent 7890B GC oven coupled to a 5977B mass spectrometer detector (Agilent Technologies, Santa Clara, USA) fitted with an MPS autosampler (Gerstel GmbH & Co. KG, Mülheim an der Ruhr, Germany), as before[53] (Supplementary Materials for a detailed protocol).

### Short chained-fatty acid (SCFA) analysis
Metabolites were prepared and derivatized following the protocol developed by Furuhashi et al.[7], with some modifications followed by analysis on an Agilent 6890B gas chromatograph (GC) oven coupled to a 5977B mass spectrometer (MS) detector (Agilent Technologies, Mulgrave, VIC, Australia) fitted with an multipurpose (MPS) autosampler (Gerstel GmbH and Co.KG, Mülheim an der Ruhr, Germany) (Supplementary Materials for a detailed protocol).

### Statistical analysis of metabolomics data
Normalised data were analysed using multivariate data analysis software SIMCA 16 (version 16, Sartorius Stedim Biotech, Umeå, Sweden). The data matrices were log-transformed to generate more symmetric distributions, and Pareto scaled for comparability across metabolites[54]. PLS-DA classification models were generated to reduce the data dimensionality and resolve the metabolite differences between HAD groups. Principal scores plots assessed how well clinical groupings could differentiate the HAD metabolome. PLS-DA is prone to over-fitting the data, and model reliability requires cross-validation[54]. Cross-validation was performed in SIMCA using $R^2X$, $R^2Y$ and $Q^2$ values along with cross-validation analysis of variance (CV-ANOVA) that determined PLS-DA model significance[55]. Models with p-values < 0.05 were deemed statistically significant. $R^2$ values greater than 0.67 was considered to have a high predictive accuracy, a range of 0.33-0.67 indicated a moderated effect, $R^2$ between 0.19 and 0.33 indicated a low effect, while $R^2$ values below 0.19 were considered unacceptable[54]. Highly disparate $R^2$ and $Q^2$ values indicated possible model over-fitting in supervised analyses[54]. Multivariate ROC-AUC analysis was performed in SIMCA to assess the performance of each PLS-DA classifier in modelling each clinical sub-group. Metabolites with variable importance in projection (VIP) scores greater than 1.0 and predictive loading values (p(corr)) greater than 0.5 and less than -0.5 were retained. Heatmaps of significant metabolites visualising the abundance of metabolites across clinical sub-groups were generated using ClustVis[56].

Statistically significant metabolites were assessed in GraphPad Prism version 8.2.1 for Windows (GraphPad Software, San Diego, California, USA) using the non-parametric Mann-Whitney U test and Kruskal-Wallis H test, FDR adjusted for multiple comparisons using the Benjamini and Hochberg method. Univariate ROC-AUC assessed the performance of putative biomarkers. An AUC cut-off of 70% was set in the current study, and biomarkers were assessed according to the following criteria: 90–100% = excellent; 80–90% = good; 70–80% = fair; 60–70% = poor[57]. Biomarkers were further assessed using the metabolite ratios to assess the relationships between metabolite elevation and depletions between biologically significant metabolite pairs.

### Reporting summary
Further information on research design is available in the Nature Portfolio Reporting Summary linked to this article.

## Data availability
The raw 16 S rRNA sequence data generated in this study have been deposited in the NCBI database under BioProject PRJNA986597, accession numbers SRR249990219, SRR24999020, SRR24999021, and SRR24999022. Sequences can be accessed at. The raw *E. faecium* sequence data generated in this study have been deposited in the NCBI database under BioProject ID PRJNA1015000, accession numbers SAMN37345311-SAMN37345366). Sequences can be accessed at: https://www.ncbi.nlm.nih.gov/biosample/37345312 https://www.ncbi.nlm.nih.gov/biosample/37345313; https://www.ncbi.nlm.nih.gov/biosample/37345314 https://www.ncbi.nlm.nih.gov/biosample/37345315; https://www.ncbi.nlm.nih.gov/biosample/37345316 https://www.ncbi.nlm.nih.gov/biosample/37345317; https://www.ncbi.nlm.nih.gov/biosample/37345318 https://www.ncbi.nlm.nih.gov/biosample/37345319; https://www.ncbi.nlm.nih.gov/biosample/37345320 https://www.ncbi.nlm.nih.

gov/biosample/37345321; https://www.ncbi.nlm.nih.gov/biosample/37345322 https://www.ncbi.nlm.nih.gov/biosample/37345323; https://www.ncbi.nlm.nih.gov/biosample/37345324 https://www.ncbi.nlm.nih.gov/biosample/37345325; https://www.ncbi.nlm.nih.gov/biosample/37345326 https://www.ncbi.nlm.nih.gov/biosample/37345327; https://www.ncbi.nlm.nih.gov/biosample/37345328 https://www.ncbi.nlm.nih.gov/biosample/37345329; https://www.ncbi.nlm.nih.gov/biosample/37345330 https://www.ncbi.nlm.nih.gov/biosample/37345331; https://www.ncbi.nlm.nih.gov/biosample/37345332 https://www.ncbi.nlm.nih.gov/biosample/37345333; https://www.ncbi.nlm.nih.gov/biosample/37345334 https://www.ncbi.nlm.nih.gov/biosample/37345335; https://www.ncbi.nlm.nih.gov/biosample/37345336 https://www.ncbi.nlm.nih.gov/biosample/37345337; https://www.ncbi.nlm.nih.gov/biosample/37345338 https://www.ncbi.nlm.nih.gov/biosample/37345339; https://www.ncbi.nlm.nih.gov/biosample/37345340 https://www.ncbi.nlm.nih.gov/biosample/37345341; https://www.ncbi.nlm.nih.gov/biosample/37345342 https://www.ncbi.nlm.nih.gov/biosample/37345343; https://www.ncbi.nlm.nih.gov/biosample/37345344 https://www.ncbi.nlm.nih.gov/biosample/37345345; https://www.ncbi.nlm.nih.gov/biosample/37345346 https://www.ncbi.nlm.nih.gov/biosample/37345347; https://www.ncbi.nlm.nih.gov/biosample/37345348 https://www.ncbi.nlm.nih.gov/biosample/37345349; https://www.ncbi.nlm.nih.gov/biosample/37345350 https://www.ncbi.nlm.nih.gov/biosample/37345351; https://www.ncbi.nlm.nih.gov/biosample/37345352 https://www.ncbi.nlm.nih.gov/biosample/37345353; https://www.ncbi.nlm.nih.gov/biosample/37345354 https://www.ncbi.nlm.nih.gov/biosample/37345355; https://www.ncbi.nlm.nih.gov/biosample/37345356 https://www.ncbi.nlm.nih.gov/biosample/37345357; https://www.ncbi.nlm.nih.gov/biosample/37345358 https://www.ncbi.nlm.nih.gov/biosample/37345359; https://www.ncbi.nlm.nih.gov/biosample/37345360 https://www.ncbi.nlm.nih.gov/biosample/37345361; https://www.ncbi.nlm.nih.gov/biosample/37345362 https://www.ncbi.nlm.nih.gov/biosample/37345363; https://www.ncbi.nlm.nih.gov/biosample/37345364 https://www.ncbi.nlm.nih.gov/biosample/37345365 The metabolomics data generated in this study are provided in Supplementary Data file 1 and the Source Data file. Source data are provided with this paper.

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

## Acknowledgements

D.L. acknowledges funding from The Australian National Health and Medical Research Council (NHMRC Ideas Grant 2002156) and Australian Research Council (ARC) Laureate Fellowship.

## Author contributions

Conceptualization, M.B., G.J., P.J., D.L.; Methodology, M.B., A.K., T.T.H.V., D.K., G.J., S.C., R.M., D.B., EP, SL, DL; Formal analysis and investigation, M.B., A.K., T.T.H.V., G.J., RM, D.B., E.P., S.L., D.L.; Resources, D.K., G.J., S.C., R.M., D.B., E.P., D.L.; Writing – original draft, MB, SL, DL; Writing – review & editing, M.B., A.K., G.J., S.C., P.J., R.M., D.B., Y.S., E.P., S.L., D.L.; Funding Acquisition, D.L.

## Competing interests

The authors declare no competing interests.

## Additional information

¹Monash Biomedicine Discovery Institute and Department of Microbiology, Monash University, Clayton, Victoria, Australia. ²Environment, Commonwealth Scientific and Industrial Research Organisation, Ecosciences Precinct, Dutton Park, Queensland, Australia. ³Department of Chemistry and Biotechnology, Swinburne University of Technology, Hawthorn, Victoria, Australia. ⁴Agriculture and Food, Commonwealth Scientific and Industrial Research Organisation, Acton, ACT, Australia. ⁵School of Science, RMIT University, Bundoora, Victoria, Australia. ⁶Department of Infectious Diseases, Monash Health, Clayton, Victoria, Australia. ⁷Department of Gastroenterology, The Queen Elizabeth Hospital, Woodville South, South Australia, Australia. ⁸These authors contributed equally: Sarah Larcombe and Dena Lyras. ✉e-mail: Dena.Lyras@monash.edu

