## [Peer Review File · Nature Communications]

REVIEWER COMMENTS

Reviewer #1 (Remarks to the Author):

1. What are the noteworthy results?

Characterization of microbiota and metabolome dynamics in CDI, including biomarkers of amino acid competition, colonisation resistance, and Stickland fermentation biomarkers that could distinguish CDI from non-CDI patients. This study offers candidate metabolic biomarkers for diagnostic development may have implications regarding CDI and vancomycin-resistant enterococci (VRE) treatment.

2. Will the work be of significance to the field and related fields? How does it compare to the established literature?

Host-microbiome interactions in the gut are not fully understood in difficult to treat bacterial infections such as *C. diff*. As the authors state, this study provides useful information on "uncovered previously unrecognised microbiota and metabolome dynamics in CDI, including biomarkers of amino acid competition, colonisation resistance, and Stickland fermentation biomarkers that could distinguish CDI from non-CDI patients. This study offers candidate metabolic biomarkers for diagnostic development may have implications regarding CDI and vancomycin-resistant enterococci (VRE) treatment." Overall, this work gives further insight into developing a potential panel of biomarkers to track and treat gut related bacterial infections.

3. If the work is not original, please provide relevant references.

Sufficient references provided.

4. Does the work support the conclusions and claims, or is additional evidence needed?

Sufficient data and discussion are provided to support the conclusions.

5. Sufficient experimental evidence is provided to support the aims of the study.

Sufficient experimental evidence and data are provided to support the aims of the study.

5. Are there any flaws in the data analysis, interpretation and conclusions? Do these prohibit publication or require revision?

There is sufficient presentation of results and discussion.

To improve upon the discussion, a summary of paragraph of key metabolites, associated pathways, and significance may be helpful.

6. Is the methodology sound? Does the work meet the expected standards in your field

Methodology is fine.

7. Is there enough detail provided in the methods for the work to be reproduced?

More detail is needed on how samples were prepared and extracted for GC/MS analysis.

Reviewer #2 (Remarks to the Author):

In this manuscript "Multi-omics analysis of hospital-acquired diarrhoeal patients reveals biomarkers of enterococcal and *Clostridioides difficile* infection" Lyras and colleagues analyze fecal samples from a clinical cohort of hospital-acquired diarrhea patients. A subset of this cohort was associated with antibiotic usage, *C. difficile* infection or presence of *Enterococcus*. The team performing 16S rRNA microbial sequencing to profile the composition of the microbiota from these samples as well as untargeted metabolomics by Gas Chromatography- Mass Spectrometry. The team generated a wealth of data in this manuscript that is of use to the infectious disease clinical and research community. However, some conclusions are clouded and difficult to interpret the significance. It is challenging to follow the main concepts being presented between figures, supplemental figures and text. Further it is unclear if the biomarkers identified, like the increased tyramine to L-tyrosine ratio observed in enterococcal-dominant patients, will be consistently observed in multiple patient cohorts. Currently the data is presented in manner that makes it difficult for the reader to distill clear takeaway messages from the dataset. This reviewer has the following critiques:

Major Critiques:

1. The manuscript provides a large dataset and multiple analyses that make the manuscript feel disjointed. The first half of the manuscript (Fig 1-2) focuses on comparing the presence or absence of *Enterococcus* while the second half of the manuscript (Fig 3-5) compares *C. difficile* infected and non infected patients plus or minus the variables of antibiotic usage, enterococcus presence. The analysis is done on the same clinical cohort dataset but grouped in multiple different ways. The manuscript clarity would benefit from focusing on one comparison condition (i.e. +/- *Enterococcus* or +/- *C. difficile*).

2. The first subsection of the Results section is reported in supplemental figures. The authors should consider moving the supplemental figure to the main portion of the text or restructuring the Results section to start with Figure 1 results.

3. The authors use a cohort of FMT donors as a control comparison. More details is needed in the Results section and main figures on the demographics of these donors and how they compare to the clinical cohort.

4. It is unclear if the FMT donor are the proper group for comparison when identifying some of the biomarkers.

5. Figure 3 conclusions that Non-antibiotic and non-enterococcal CDI is metabolically distinct from the enterococcal CDI" is not supported. The heatmap in figure 3A does not show a distinct metabolite profile and the PLS-DA plot in Figure 3B seems to show the FMT donors are distinct from all other groups, but the rest of the groups are not distinguishable.

Minor Critiques:

1. Results subsection "Enterococcal-dominant AAD was significantly depleted in SCFAs" Ln 214-223 seems to be referring to supplementary figure 5, not suppl. Fig 6. Further the data in Suppl Fig. 5 seems to show that both enterococcal dominant and negative AAD have significant less SCFAs.

2. Ln 303-304 "5-aminovaleric acid/proline ratios were reduced in non-CDI patients compared to their non-CDI counterparts who shared a similarly elevated..." is a confusing sentence and doesn't seem to be supported by the data in Fig 4C

3. Fig 4C and Fig 4 legend mislabeled +Ent-CDI combination

Reviewer #3 (Remarks to the Author):

The manuscript by Bosnjak et al describes the gut microbial (both taxonomic and metabolomic) differences within a patient cohort with antibiotic-associated diarrhoea (AAD). While *C. difficile* infection (CDI) remains the largest cause of AAD, not all AAD is directly explained by *C. difficile*. Coupled with the fact that antibiotic use carries the risk of gut microbiota perturbation and increases in gut dominance by

antibiotic-resistant organisms (such as Enterococcus), identifying microbial perturbation following antibiotic use is a timely study. The authors further dichotomized their AAD data set (initially CDI vs. non-CDI) into Enterococcus(Ent) (+) or (-) groups, and identified that most metabolomic and taxonomic differences were present between Ent(+) and (-) groups, independent from CDI. From both 16S rRNA gene-based sequencing and metabolomics, the authors observed that most differences were actually explained by the presence (or rather, dominance) of Enterococcus, independent CDI. All groups were significantly distinct from a control FMT cohort representing a 'healthy' cohort, validating previous similar studies in humans. Overall, the data supports the conclusions, namely that Enterococcus-C. difficile, and perhaps even Enterococcus-dominated microbiota, are distinct from other types of AAD, including CDI. The manuscript is comprehensive and well-written. Specific comments below:

- While the data itself support the author conclusions, the abstract is a bit misleading. The abstract mostly summarizes some of the CDI/non-CDI comparisons, and mostly neglects the data surrounding Ent-/+ comparisons. Given that the focus of the results is mainly on ENT-/+, perhaps the abstract could focus more on the Ent comparisons?

- The manuscript has some mismatches between figure references and the actual figures (mostly within supplemental). See below:

o Figure S1 has a part D, but not included in the manuscript?

o Figure S2: AB and CD panels are switched in the text vs. Figure S2

o Lines 218-223: the SCFA amounts refer to Figure S5, not S6 (as in text)

o (line 281/2) The referenced figures (Figure 3C-E) are not heatmaps, and there is no Figure 3E

- The microbially distinct profiles of the Ent+ AAD patients vs others is striking and overall, the data is supportive of this idea. That said, it is possible that stratifying the data based on > 25% Enterococcus dominance is artificially separating these samples. Alpha- and beta-diversity calculations will be impacted by anything that represents > 25% of the community, and both the decreased diversity and structural separation could be explained by this alone. Perhaps this should be rephrased that the samples with decreased diversity are dominated by Enterococcus? Alternately, to assess the rest of the community, the same calculations could be done after removing all Enterococcus sequences.

- (lines 255-262) What features exactly demonstrated that the -AAD+CDI, -Ent+CDI metabolomes were overall "less perturbed" than their CDI+ counterparts? From the heatmap (Figure 3A), they seem similar to each other, different from their CDI+ counterparts, but also quite different from the FMT group?

- Methods clarification:

- Were the Shannon index and beta diversity calculations done on OTU abundances?

- The raw sequencing (both 16S rRNA and genomic) and metabolomic data should be deposited in a publicly available database.

Response to Reviewers for Manuscript NCOMMS-23-24210

Multi-omics analysis of hospital-acquired diarrhoeal patients reveals biomarkers of enterococcal proliferation and *Clostridioides difficile* infection

Note that reviewer comments are in black text and author responses are in blue text. Line numbers refer to the document containing tracked changes (with track changes showing).

Note that a Data Availability Statement has now been added to the main manuscript (Lines 36-39).

Reviewer #1 (Remarks to the Author):

1. What are the noteworthy results?

Characterization of microbiota and metabolome dynamics in CDI, including biomarkers of amino acid competition, colonisation resistance, and Stickland fermentation biomarkers that could distinguish CDI from non-CDI patients. This study offers candidate metabolic biomarkers for diagnostic development may have implications regarding CDI and vancomycin-resistant enterococci (VRE) treatment.

2. Will the work be of significance to the field and related fields? How does it compare to the established literature?

Host-microbiome interactions in the gut are not fully understood in difficult to treat bacterial infections such as *C. diff*. As the authors state, this study provides useful information on "uncovered previously unrecognised microbiota and metabolome dynamics in CDI, including biomarkers of amino acid competition, colonisation resistance, and Stickland fermentation biomarkers that could distinguish CDI from non-CDI patients. This study offers candidate metabolic biomarkers for diagnostic development may have implications regarding CDI and vancomycin-resistant enterococci (VRE) treatment." Overall, this work gives further insight into developing a potential panel of biomarkers to track and treat gut related bacterial infections.

3. If the work is not original, please provide relevant references.

Sufficient references provided.

4. Does the work support the conclusions and claims, or is additional evidence needed?

Sufficient data and discussion are provided to support the conclusions.

5. Sufficient experimental evidence is provided to support the aims of the study.

Sufficient experimental evidence and data are provided to support the aims of the study.

5. Are there any flaws in the data analysis, interpretation and conclusions? Do these prohibit publication or require revision?

There is sufficient presentation of results and discussion.

To improve upon the discussion, a summary of paragraph of key metabolites, associated pathways, and significance may be helpful.

- We agree. However, this was addressed in the original manuscript in lines 457-473. Thus, we have not altered the manuscript as the reviewer's comment was already addressed. This paragraph reads as follows: "In the current study, several C. difficile-specific biomarkers derived from Stickland fermentation of branched-chain amino acids could differentiate CDI patients from non-CDI patients. These results suggest that in human CDI, C. difficile may preferentially catabolise amino acids over sugars during infection, which is consistent with the hypothesis that the availability of faecal amino acids enhances CDI susceptibility 27, 40. Specifically, 4-MPA, isobutyrate, isovalerate, 5-aminovaleric acid and desaminotyrosine

formed putative CDI biomarkers. However, the contributions of these Stickland products may be due to unaccounted-for taxa as the presence of other Stickland fermenting clostridia, including Clostridium bifermentans, Clostridium botulinum, Clostridium sporogenes, Paraclostridium sordellii and Peptostreptococcus anaerobius, were not detected in the current study 41. Similarly, while C. difficile can produce small amounts of desaminotyrosine during its growth phase, desaminotyrosine has not been previously associated with CDI 27. Instead, elevated desaminotyrosine has been associated with microbiota response to infection and inflammation by commensal Stickland fermenting bacteria such as C. orbiscindens, which were not detected in the current study 42, 43. However, detection of desaminotyrosine and other Stickland products may also reflect C. difficile metabolic diversity in response to microbiota competition for amino acids.

6. Is the methodology sound? Does the work meet the expected standards in your field

Methodology is fine.

7. Is there enough detail provided in the methods for the work to be reproduced?

More detail is needed on how samples were prepared and extracted for GC/MS analysis.

- *A description for metabolomics extraction and derivatisation has been added to the supplementary materials section, see pages 5-9. This is in addition to details in the methods section of the main manuscript (Lines 621-639).*

Reviewer #2 (Remarks to the Author):

In this manuscript “Multi-omics analysis of hospital-acquired diarrhoeal patients reveals biomarkers of enterococcal and Clostridioides difficile infection” Lyras and colleagues analyze fecal samples from a clinical cohort of hospital-acquired diarrhea patients. A subset of this cohort was associated with antibiotic usage, C. difficile infection or presence of Enterococcus. The team performing 16S rRNA microbial sequencing to profile the composition of the microbiota from these samples as well as untargeted metabolomics by Gas Chromatography- Mass Spectrometry. The team generated a wealth of data in this manuscript that is of use to the infectious disease clinical and research community.

However, some conclusions are clouded and difficult to interpret the significance. It is challenging to follow the main concepts being presented between figures, supplemental figures and text. Further it is unclear if the biomarkers identified, like the increased tyramine to L-tyrosine ratio observed in enterococcal-dominant patients, will be consistently observed in multiple patient cohorts. Currently the data is presented in manner that makes it difficult for the reader to distill clear takeaway messages from the dataset.

- *We agree that it may be difficult to follow the main concepts. To make concepts and takeaway messages easier to follow we have moved key figures to the main text. Specifically, we have addressed the following:*
- *To better highlight the link between antibiotics, low diversity enterococcal-dominant microbiota and putative tyramine/tyrosine biomarkers, we have completed the following:*
 - o *Moved ‘Supplementary Table 3 Patient demographics and antibiotic usage’ from the supplementary materials into the manuscript and relabelled it as ‘Table 1. Patient demographics and antibiotic usage’. (Line 702)*
 - o *Moved ‘Supplementary Figure 2. Extended antibiotic exposure and combination antibiotic therapy was associated with a microbiota depleted in Lachnospiraceae and dominated by Enterococcaceae’ from the supplementary materials into to manuscript and relabelled it as ‘Figure 1. Extended antibiotic exposure and combination antibiotic therapy was associated with a microbiota depleted in Lachnospiraceae and dominated by Enterococcaceae’. (Line 704)*
 - o *Combined ‘Supplementary Figure 3. Summary of patient cohorts stratified by Enterococcal dominance’ and ‘Supplementary Figure 4. Enterococcal-dominant AAD*

formed a microbially distinct subset of AAD' and moved into the manuscript and relabelled as 'Figure 3. Low diversity enterococcal-dominant AAD formed a microbially distinct subset of AAD'. (Line 725)

- To better highlight the metabolite differences (specifically Stickland biomarkers) between CDI and non-CDI patients with and without concomitant enterococcal proliferation, we have completed the following:

- o In addition to figures relating to L-proline, L-leucine and L-valine Stickland biomarkers, we moved 'Supplementary Figure 7. Non-enterococcal CDI was associated with by-products of tyrosine Stickland fermentation' from the supplementary materials into the manuscript and relabelled as 'Figure 8. Non-enterococcal CDI was associated with by-products of tyrosine Stickland fermentation' (Line 800)*

- Regarding the putative biomarkers identified, we feel the size of the patient cohort was appropriate to indicate trends in biomarkers. Whether the same trend can be observed in similar cohorts remains to be seen. However, clinical multiomics studies of diarrhoeal and CDI cohorts are not extensive, particularly those characterising a spectrum of antibiotic associated dysbiosis. In relation to the relationship between tyramine and enterococcal proliferation, this is not a relationship that has been a feature of clinical studies, however, enterococcal proliferation and tyramine biomarkers have been noted among patients with Parkinson's disease undergoing L-dopa treatment (Rekdal et al Science. 2019 364(6445): eaa06323 (Ref 1). We did not have access to Parkinson's treatment data in our cohort but it would be interesting to see if Parkinson's treatment could be a contributing factor in future studies.

Similarly, C. difficile Stickland biomarkers have been poorly investigated in clinical settings and this study to our knowledge now forms one of two studies that have pursued this line of investigation. The study by Robinson et al identified different Stickland biomarkers (Robinson et al J Clin Invest 2019 Aug 12;129(9):3792-3806 (Ref 2) in a clinical study of CDI metabolomes among diarrhoeal patients). Therefore, it also remains to be seen whether the Stickland biomarkers presented in the current study will be replicated.

This reviewer has the following critiques:

Major Critiques:

1. The manuscript provides a large dataset and multiple analyses that make the manuscript feel disjointed. The first half of the manuscript (Fig 1-2) focuses on comparing the presence or absence of Enterococcus while the second half of the manuscript (Fig 3-5) compares C. difficile infected and non infected patients plus or minus the variables of antibiotic usage, enterococcus presence. The analysis is done on the same clinical cohort dataset but grouped in multiple different ways. The manuscript clarity would benefit from focusing on one comparison condition (i.e. +/- Enterococcus or +/- C. difficile).

- Although we could have performed the comparison in a number of different ways, we feel that by doing it in the way that we have we bring forth multiomics insights and characteristics that relate to a significant clinical problem, that being antibiotic associated diarrhoea, bacterial overgrowth in a low diversity microbiota, and CDI risk. These outcomes are important because while concomitant enterococcal proliferation with CDI has been observed, there have been no studies, to our knowledge, that have explored these ecosystems and interactions within the context of metabolomic activity. We have sought to understand whether within the disorderliness and multiple variables impacting the human gut microbiota, findings from in vivo studies could be observed and help us understand those relationships. Overlaying the enterococcal story with the C. difficile story is important as both are strongly associated with antibiotic usage and need to be investigated together.*

2. The first subsection of the Results section is reported in supplemental figures. The authors should consider moving the supplemental figure to the main portion of the text or restructuring the Results section to start with Figure 1 results.

- *The results section now starts with the patient demographics table 1 (Line 702) and key figures that were in Supplementary (previously Supp Figures 2-4 & 7) have been reincorporated in the main paper in results (now Figures 1, 3 & 8 – Lines 704, 725 and 800) which we feel makes the story clearer.*

3. The authors use a cohort of FMT donors as a control comparison. More detail is needed in the Results section and main figures on the demographics of these donors and how they compare to the clinical cohort.

- *The patient demographics have been included (Table 1, Line 702) and demographic data for FMT donors is in the Supplementary Materials, specifically in the FMT donor specimen inclusion criteria section, Page 2-3, lines 27-35. The information is also in the manuscript (Lines 145-155).*

4. It is unclear if the FMT donor are the proper group for comparison when identifying some of the biomarkers.

- *FMT donors are very healthy. Only 3% of potential applicants to the donor screening program pass screening. They must have no active medical problems and be on no medication and not have risk factors for transmissible disease. In addition, they must have normal blood and stool tests on a wide range of analyses. These are a good group in the sense of being “healthy” but may not be ideal as they will be a much younger cohort than a hospitalised cohort or cohort with CDI.*
- *A rationale for use of FMT donor samples has been included in the Supplementary Materials (pages 2-3, Lines 17-35). This states “For comparison, faecal samples were collected from healthy donors recruited for faecal microbiota transplantation (FMT) treatment of CDI. FMT is a unique treatment approach for recurrent CDI that aims to restore the commensal gut microbiota and in turn re-establish colonisation resistance to inhibit the growth of *C. difficile*. While high microbial diversity coupled with high proportions of SCFA and secondary bile acid producing bacteria are considered to be the key to successful FMT, little is known clinically, about the role of amino acids and their fermentation products in *C. difficile* decolonisation. Most recently, studies have shown that FMT restoration of commensal microbiota increased competition for these preferred amino acids where species such as *Clostridium sardiniense*, with similar nutritional requirements as *C. difficile*, deplete amino acids in the gut to provide substantial protection against CDI 26.” “The health status and eligibility of all donors were screened following BiomeBank’s screening protocols that included an interview, medical assessment, blood, and stool screening. All donor samples in the study were obtained from 12 female and 8 male individuals, a smaller male cohort (40.0% male) compared to CDI patients (54.5% male) and non-CDI patients (47.4% male). FMT donors were between the ages of 18 and 65 46 with a total age of 649 years and an average of 32.45 years. While the average age of FMT donors was significantly lower than the CDI (75 years, range 55-83) and non-CDI (68 years, range 52-78) median age, we chose FMT donors purposefully as a comparison group in order to assess the microbiota and metabolomes of CDI and non-CDI patients against FMT donors who are medically assessed as healthy and are actively recruited to treat recurrent and severe CDI.”*

5. Figure 3 conclusions that Non-antibiotic and non-enterococcal CDI is metabolically distinct from the enterococcal CDI” is not supported. The heatmap in figure 3A does not show a distinct metabolite profile and the PLS-DA plot in Figure 3B seems to show the FMT donors are distinct from all other groups, but the rest of the groups are not distinguishable.

- *Thank you for your comments and apologies for the error. Figure 3 has been moved and renamed for accuracy and is now Figure 5 (line 750). We acknowledge that this caption is confusing and erroneous; thus, we have re-worded the caption title to better reflect the data – “Figure 5. Non-antibiotic and non-enterococcal CDI metabolomes shared a reduction in sugars and amino acids compared to enterococcal CDI and non-CDI metabolomes (Line 742). Additionally, we have corrected the reference to the figure in the results section (Lines 299-301) to state “-AAD+CDI and -Ent+CDI metabolomes were associated with reduced*

sugars, sugar alcohols and amino acids compared to less perturbed than their non-CDI counterparts”

Minor Critiques:

1. Results subsection “Enterococcal-dominant AAD was significantly depleted in SCFAs” Ln 214-223 seems to be referring to supplementary figure 5, not suppl. Fig 6. Further the data in Suppl Fig. 5 seems to show that both enterococcal dominant and negative AAD have significant less SCFAs.

- *We apologise for this error. This has been corrected and the correct Supp Figure has been referenced in the text (new Supp Fig 1, Lines 315, 316, 323 & 326).*

- *Thank you for your feedback. You are correct - compared to FMT donors, both enterococcal dominant and negative AAD have significantly less SCFAs. However, enterococcal dominant AAD also has significantly less SCFAs (specifically propionate and butyrate) compared to enterococcal negative AAD and it was this difference we were most interested in communicating. Supplementary Figure 5 has been relabelled as Supplementary Figure 1 (page 22 of the supplementary materials) and we have changed the caption from “Enterococcal-dominant AAD was enriched in metabolites across several classes and depleted in short-chain and branched-chain fatty acids, forming a metabolically distinct subset of AAD” to “Enterococcal-dominant AAD was enriched in metabolites across several classes and depleted in propionate and butyrate, forming a metabolically distinct subset of AAD” (see page 24 of supplementary materials).*

2. Ln 303-304 “5-aminovaleric acid/proline ratios were reduced in non-CDI patients compared to their non-CDI counterparts who shared a similarly elevated...” is a confusing sentence and doesn't seem to be supported by the data in Fig 4C

- *We apologise for the lack of clarity. This sentence has now been changed to be concise and clear, see lines 352-356. “Mean 5-aminovaleric acid/proline ratios were reduced in non-CDI patients compared to their CDI counterparts but the differences in 5-aminovaleric acid/proline ratios between CDI groups and their non-CDI counterparts were not statistically significant (Figure 6C).”*

3. Fig 4C and Fig 4 legend mislabeled +Ent-CDI combination

We apologise for the error. Fig 4 is now Fig 6. In both Fig 6A and 6C, the axes labels in the figures have been edited from the originally mislabelled +-Ent-CDI to +Ent-CDI i.e. the + - prefixes have been corrected. See line 771.

Reviewer #3 (Remarks to the Author):

The manuscript by Bosnjak et al describes the gut microbial (both taxonomic and metabolomic) differences within a patient cohort with antibiotic-associated diarrhoea (AAD). While *C. difficile* infection (CDI) remains the largest cause of AAD, not all AAD is directly explained by *C. difficile*. Coupled with the fact that antibiotic use carries the risk of gut microbiota perturbation and increases in gut dominance by antibiotic-resistant organisms (such as *Enterococcus*), identifying microbial perturbation following antibiotic use is a timely study. The authors further dichotomized their AAD data set (initially CDI vs. non-CDI) into *Enterococcus* (Ent) (+) or (-) groups, and identified that most metabolomic and taxonomic differences were present between Ent(+) and (-) groups, independent from CDI. From both 16S rRNA gene-based sequencing and metabolomics, the authors observed that most differences were actually explained by the presence (or rather, dominance) of *Enterococcus*, independent CDI. All groups were significantly distinct from a control FMT cohort representing a ‘healthy’ cohort, validating previous similar studies in humans. Overall, the data supports the conclusions, namely that *Enterococcus-C. difficile*, and perhaps even *Enterococcus*-dominated microbiota, are distinct from other types of AAD, including CDI. The manuscript is comprehensive and well-written. Specific comments below:

1. While the data itself support the author conclusions, the abstract is a bit misleading. The abstract mostly summarizes some of the CDI/non-CDI comparisons, and mostly neglects the data surrounding Ent-/+ comparisons. Given that the focus of the results is mainly on ENT-/+ , perhaps the abstract could focus more on the Ent comparisons?

- *The abstract has been edited to make clearer the link between antibiotic treatment and Enterococcus and Enterococcus proliferation and CDI. Specifically, we have added the following “Notably, extended antibiotic treatment was associated with enterococcal proliferation (mostly vancomycin-resistant Enterococcus faecium) coupled with putative biomarkers of enterococcal tyrosine decarboxylation. We also uncovered previously unrecognized metabolome dynamics associated with concomitant enterococcal proliferation and CDI, including biomarkers of Stickland fermentation biomarkers and amino acid competition that could distinguish CDI from non-CDI patients.” See lines 52-56.*

2. The manuscript has some mismatches between figure references and the actual figures (mostly within supplemental). See below:

o Figure S1 has a part D, but not included in the manuscript?

- *Fig S1 is now Fig 3. Part D of the figure legend has been removed. Lines 725-734*

o Figure S2: AB and CD panels are switched in the text vs. Figure S2

- *Fig S2 is now Fig 1 and the panel mix up has been corrected. Lines 705-711*

o Lines 218-223: the SCFA amounts refer to Figure S5, not S6 (as in text)

- *Fixed - reference made to the correct Supp Figure - new Supplementary Fig 1*

o (line 281/2) The referenced figures (Figure 3C-E) are not heatmaps, and there is no Figure 3E

- *Removed the word heatmaps and removed reference to Figure 3E. It is now Figure 5D (line 763)*

3. The microbially distinct profiles of the Ent+ AAD patients vs others is striking and overall, the data is supportive of this idea. That said, it is possible that stratifying the data based on > 25% Enterococcus dominance is artificially separating these samples. Alpha- and beta-diversity calculations will be impacted by anything that represents > 25% of the community, and both the decreased diversity and structural separation could be explained by this alone. Perhaps this should be rephrased that the samples with decreased diversity are dominated by Enterococcus? Alternately, to assess the rest of the community, the same calculations could be done after removing all Enterococcus sequences.

- *Thank you for your comments. We agree that stratifying the data based on > 25% Enterococcus dominance could be viewed as artificially separating these samples. Analysis of the PCOA (Figure 3C – line 734) for all HAD patients revealed that patients who were characterised by a low-diversity microbiota with Enterococcus exceeding 25% of the total microbiota clustered to the right of the PCOA and were strongly correlated with combination and extended antibiotic treatment. In addition, we considered that the horse shoe shape observed in the PCOA (Figure 3C – line 734), an ordination structure that has been studied by Morton et al (2017) (Morton et al. 2017. mSystems Vol 2, No. 1)(Ref 3), showed clear separation of the communities based on diversity, where low diversity samples corresponded to enterococcal proliferation following antibiotic treatment.*
- *Although we would happily re-analyse and provide additional data following the exclusion of enterococcal samples, we no longer have access to the Calypso microbiota package (Zakrzewski, M et al. Bioinformatics. 2017 Mar 1; 33(5): 782–783) (Ref 4). The package was withdrawn without notice from web servers in 2021 and the corresponding authors no longer responded to our requests. We feel that the data as presented is still a valuable resource to the research community.*
- *We have rephrased sections within the manuscript to indicate decreased diversity samples were dominated by Enterococcus as suggested*

4. (lines 255-262) What features exactly demonstrated that the -AAD+CDI, -Ent+CDI metabolomes were overall “less perturbed” than their CDI+ counterparts? From the heatmap (Figure 3A), they seem similar to each other, different from their CDI+ counterparts, but also quite different from the FMT group?

- *We apologise for the error. Yes, -AAD+CDI, -Ent+CDI metabolomes are similar to each other, and the caption and sentence should read ‘different from their non-CDI counterparts’ rather than ‘CDI counterparts’. The heatmap (Figure 3A now Figure 5A – line 753) shows AAD+CDI, -Ent+CDI groups are similar to each other and the features that signified that these metabolomes were “less perturbed” than their non-CDI counterparts was they were on average lower in sugars and amino acids. The sentence has been amended and now reads as follows “-AAD+CDI and -Ent+CDI metabolomes were associated with reduced sugars, sugar alcohols and amino acids compared to their non-CDI counterparts” (Lines 299-303)*

Methods clarification:

5. Were the Shannon index and beta diversity calculations done on OTU abundances?

- *Yes, they were. The sentence has now been made clearer (Lines 592-596). “Using OTU abundances, alpha diversity was determined by calculating the Shannon diversity index and statistical significance was determined using the non-parametric Mann-Whitney U and Kruskal-Wallis H tests. Beta diversity calculations from OTU abundances were determined ...”*

6. The raw sequencing (both 16S rRNA and genomic) and metabolomic data should be deposited in a publicly available database.

- *The 16S rRNA sequence data are publicly available from the NIH Sequence Read Archive (SRA) under BioProject PRJNA986597, accession numbers SRR24999020, SRR24999021, SRR24999022 and SRR249990213. This is now detailed in the methods section, lines 584-586 of the main manuscript. The E. faecium sequence data is available from the NCBI database under BioProject ID PRJNA1015000, accession numbers SAMN37345311-SAMN37345366. This is now detailed in the methods section, lines 618-619 of the main manuscript. An excel sheet and word document will be included as separate attachments as part of supplementary materials.*

REVIEWERS' COMMENTS

Reviewer #2 (Remarks to the Author):

In the revised manuscript submitted by Lyras and colleagues, the authors have made significant improvements to the structure and layout of the manuscript. The figures are now presented in a manner for the reader to follow. The manuscript has two main focuses. AAD in the presence or absence of enterococcus and with vs without C. difficile infection. By determining the ratios of various metabolites (tyramine/tryosine for enterococcus and 5-aminovaleric/proline for C. diff + enterococcus) the group has identified potential biomarkers that could predict etiology of antibiotic-associated diarrhoea for some hospitalized patients. While these biomarkers will need to be supported and verified by independent cohorts, this work represents a step towards defining the metabolite profile of these patients. The authors address all of this Reviewers' concern.

Minor suggestion:

1. Include a flow chart for the division of the C. diff patients subsets. Similar to the flow chart in Fig 3A